# Potency assessment of IFNγ-producing SARS-CoV-2-specific T cells from COVID-19 convalescent subjects

Rubén Gil-Bescós[1], Ainhoa Ostiz[1], Saioa Zalba[2], Ibai Tamayo[3,4,5], Eva Bandrés[6], Elvira Rojas-de-Miguel[1], Margarita Redondo[2], Amaya Zabalza[1,2], Natalia Ramírez[1]

The development of new therapies for COVID-19 high-risk patients remains necessary to prevent additional deaths. Here, we studied the phenotypical and functional characteristics of IFN-γ producing-SARS-CoV-2-specific T cells (SC2-STs), obtained from 12 COVID-19 convalescent donors, to determine their potency as an off-the-shelf T cell therapy product. We found that these cells present mainly an effector memory phenotype, characterized by the basal expression of cytotoxicity and activation markers, including granzyme B, perforin, CD38, and PD-1. We demonstrated that SC2-STs could be expanded and isolated in vitro, and they exhibited peptide-specific cytolytic and proliferative responses after antigenic re-challenge. Collectively, these data demonstrate that SC2-STs can be a suitable candidate for the manufacture of a T cell therapy product aimed to treat severe COVID-19.

## Introduction

Since its discovery in Wuhan, China, in December 2019, the Coronavirus disease (COVID-19), caused by the Severe Acute Respiratory Syndrome Coronavirus 2 (SARS-CoV-2), has been the cause of the greatest pandemic outbreak in the 21st century. About 587 million cases and 6.4 million deaths (data updated as of 17 August 2022) (World Health Organization, 2022) have scourged the global population.

Clinical manifestations of SARS-CoV-2 infection range from asymptomatic infection to acute pneumonia, which may require invasive mechanical ventilation in 2–16% of the hospitalized patients, depending on the virus variant that caused the infection (Karagiannidis et al, 2020; Tzotzos et al, 2020; Avdeev et al, 2021; Maslo et al, 2022). Elderly people and patients with preexisting comorbidities, such as hypertension, obesity or diabetes mellitus (Ng et al, 2021), are more susceptible to suffer severe infection, subsequent hospitalization, intensive care unit admission or death.

Patients with severe and critical disease by SARS-CoV-2 show a hyper-inflammation state caused by an exacerbated immune system response to the virus. Specifically, a significant increase in pro-inflammation biomarkers such as C-reactive protein, IL-6, IL-10, and IL-2R amongst others (Gong et al, 2020; Mahat et al, 2021) characterizes this aggravated immune reaction. This so called "cytokine storm" is associated with a drastic reduction and a high degree of exhaustion of T, B, and NK subpopulations (Jiang et al, 2020; Xu et al, 2020; Lee et al, 2021). These types of responses are associated with an increased severity of the disease and mortality. Although some medications such as the antiviral remdesivir (Beigel et al, 2020) and the glucocorticoid dexamethasone (RECOVERY Collaborative Group, 2020) have exhibited a reduction in mortality and time of recovery in severe hospitalized patients, the available treatments for ongoing severe COVID-19 patients are still limited, and the development of new therapies for high-risk patients remains necessary.

Numerous studies have shown that convalescent subjects and vaccinated individuals develop strong virus-specific responses, including neutralizing antibodies and memory T lymphocytes (Notarbartolo et al, 2021; Painter et al, 2021; Sandberg et al, 2021). Whereas antibody titer slowly wanes after SARS-CoV-2 infection (Levin et al, 2021; Xiao et al, 2021), being also probably less effective against variants of concern (VOCs) like Omicron (Planas et al, 2022), virus-specific T cell responses are maintained on the convalescent phase of the infection (Bilich et al, 2021; Dan et al, 2021) and its effectivity has been previously proven against such VOCs (Keeton et al, 2022; Redd et al, 2022). These data are endorsed by studies where immune response to the precursor SARS-CoV-1 is analysed, showing that convalescent individuals maintained detectable memory T cells more than 11 yr after the infection (Ng et al, 2016). Taking that into consideration, adoptive immunotherapy with SARS-CoV-2-specific T cells (SC2-STs) emerge as an interesting therapeutic strategy for immunosuppressed severe COVID-19 patients.

[1]Oncohematology Research Group, Navarrabiomed, University Hospital of Navarra, Public University of Navarra, Navarra Medical Research Institute (IdiSNA), Pamplona, Spain    [2]Hematology and Hemotherapy Department, University Hospital of Navarra, IdiSNA, Pamplona, Spain    [3]Unit of Methodology, Navarrabiomed, University Hospital of Navarra, Public University of Navarra, IdiSNA, Pamplona, Spain    [4]Red de Investigación en Servicios Sanitarios y Enfermedades Crónicas (REDISSEC), Pamplona, Spain    [5]Red de Investigación en Cronicidad, Atención Primaria y Promoción de la Salud (RICAPPS), Pamplona, Spain    [6]Immunology Service, University Hospital of Navarra, IdiSNA, Pamplona, Spain

Correspondence: natalia.ramirez.huerto@navarra.es; amaya.zabalza.sanmartin@navarra.es

Adoptive immunotherapy with virus-specific T cells (VSTs) obtained from an immunocompetent donor has already been used to accelerate the immune reconstitution and as prophylaxis to prevent viral diseases in patients undergoing allogenic hematopoietic stem cell transplantation, mostly CMV, EBV and adenovirus infections (Micklethwaite et al, 2007; Blyth et al, 2013; Sukdolak et al, 2013). However, until the appearance of COVID-19, the number of clinical studies where this therapeutic product had been used for the treatment of patients with respiratory infections was limited. Currently, several phase I and II clinical trials have been registered (NCT04457726, NCT04578210, NCT04765449, NCT04742595, NCT04896606) for the evaluation of the efficacy of SC2-STs as an advanced treatment of severe COVID-19 patients.

Different procedures have been implemented to isolate memory VSTs, either indirectly (allodepletion of CD45RA$^+$ naïve T cells) or directly (MHC multimer technologies and T cell activation methodology) (Sutrave et al, 2017). The production of the pro-inflammatory cytokine IFN-γ by both CD4$^+$ and CD8$^+$ memory T cell subsets after being stimulated with viral-specific peptides have also been used as a tool to obtain VSTs. The Magnetic-Activated Cell Sorting (MACS) technology and the Cytokine Capture System allow obtaining these IFN-γ-producing memory T cells (Campbell et al, 2011). Moreover, recent studies have demonstrated that it is feasible to isolate SC2-STs cells from systemic circulation with high performance using this last approach (Cooper et al, 2021; Ferreras et al, 2021). Nevertheless, hardly any studies have validated the functional potential of IFN-γ-producing SC2-STs.

In the present study, we performed a comprehensive phenotypical and functional characterization of the SARS-CoV-2-specific activated T cell subpopulation, obtained from convalescent donors recovered from mild SARS-CoV-2 primary infection. Our research is focused on the evaluation and validation of the effector function of these immunocompetent cells to endorse an off-the-shelf T cell therapy for the successful treatment of patients with severe SARS-CoV-2-associated pneumonia.

# Results

### IFN-γ-producing SC2-STs present an effector memory phenotype prevalence and exhibit basal expression of cytolytic and activation markers

12 donors that were previously diagnosed of mild COVID-19 and fulfilled the inclusion criteria were recruited. Clinical characteristics of the subjects are listed in Table 1; and the study design is represented in Fig 1. The median age of subjects was 27 ± 2 yr and 7 of them (58.3%) were female. All donors showed some symptomatology associated to mild COVID-19, with cough (83.3%), fever (41.7%), and fatigue (33.3%) being the most common symptoms. None of the donors received any kind of COVID-19-specific treatment. The number of days post symptoms onset (DPSO) to blood extraction was extremely wide, with a median value of 181.5 ± 323.7 d, and three subjects (25%) experienced a documented SARS-CoV-2 reinfection between primary infection and blood extraction day. Most donors had been vaccinated with at least the first dose of an

mRNA-based vaccine (83%), either mRNA-1273 or BNT162b2, before blood extraction. The primary infection of convalescent donors was most probably caused by different virus variants, including the ancestral virus and the alpha, delta, and omicron VOCs.

First, we assessed the IFN-γ production by freshly isolated PBMCs from convalescent donors after stimulation with SARS-CoV-2 peptides. 75 percent of donors (9/12) met the requirements mentioned in the study design section (Fig 2), and were considered valid for their future characterization. Donors 2, 6, and 10 were excluded from further analysis (Fig 3A). Interestingly, 2/3 excluded donors presented HLA-A*03, whereas none of the included donors had it in their haplotype ($P$ = 0.045, Fisher's Exact test). No significant differences were found between excluded and included donors with other HLA molecules. Upon stimulation, suitable donors exhibited a significant increase in the percentage of IFN-γ-producing CD3$^+$ cells (0.0196% ± 0.0105%; $P$ = 5.9 × 10$^{-3}$), compared with the unstimulated control (0.0052% ± 0.0027%; Fig 3B). The amount of SC2-STs in donors that met the requirements did not change in relation to the time from the infection, with no correlation found between the percentage of these cells and DPSO ($P$ = 0.087; Fig S1). In addition, virtually the same DPSO was observed between subjects that showed sufficient CD3$^+$IFN-γ$^+$ to generate SC2-STs and those who did not (280.1 ± 247.2 versus 253.7 ± 162.7, respectively; $P$ = 0.87). Regarding to sex-specific responses, we found no evidence of sex-related differences ($P$ = 0.28; Fig S2).

Next, the phenotypical characterization of IFN-γ-producing cells was carried out by Intracellular Cytokine Staining after stimulation with SARS-CoV-2 peptides, in the presence of Brefeldin A and Monensin for protein transport inhibition. Representative flow cytometry plots are shown in Fig 4A. Similar to what was observed in the IFN-γ secretion assay, the percentage of CD3$^+$IFN-γ$^+$ in the Intracellular Cytokine Staining assay increased from 0.0022% ± 0.0013% in the unstimulated control to 0.0158% ± 0.0098% ($P$ = 4.2 × 10$^{-3}$) after SARS-CoV-2 peptide stimulation. Perforin and granzyme B expression was found in all donors SC2-STs, with higher expression of both cytolytic molecules in the CD8$^+$IFN-γ$^+$ subset compared with CD4$^+$IFN-γ$^+$ cells (Fig 4B). Granzyme B was 1.8 times more abundant in CD8$^+$IFN-γ$^+$ than their CD4$^+$ counterpart, and showed high variability between donors in both of these subsets. Perforin expressing cell presence was 8.3 times higher in CD8$^+$IFN-γ$^+$, with only 5 (55.5%) donors showing detectable CD4$^+$IFN-γ$^+$perforin$^+$ cells. The expression of the surface marker FasL in SC2-STs was also measured. Contrasting with granzyme B and perforin, specific cells exhibited lower expression of FasL$^+$, and they were more abundant in the CD4$^+$IFN-γ$^+$ subpopulation. CD8$^+$IFN-γ$^+$FasL$^+$ cells were detected in only two of the selected donors (Fig 4B).

CD38 and PD-1 surface proteins were present in the SC2-STs from all studied subjects. The expression of CD38 was low in both CD4$^+$IFN-γ$^+$ and CD8$^+$IFN-γ$^+$ cells, and showed similar expression levels in both compartments. PD-1 was more abundant than CD38, showed great variability between donors, and was dominated by CD4$^+$ IFN-γ$^+$PD-1$^+$, being 5.1 times higher than CD8$^+$IFN-γ$^+$PD-1$^+$. CD3$^+$IFN-γ$^+$Ki-67$^+$ was found in 6 (66.7%) donors, in very low quantities. This marker was expressed mostly in CD8$^+$ IFN-γ$^+$ cells, and only one donor showed a minimum detectable number of CD4$^+$IFN-γ$^+$Ki-67$^+$ cells (Fig 4B).

**Table 1. SARS-CoV-2 convalescent donors.**

| Donor id | Sex | Age | HLA-I | Diagnosis | PCR ct value | Time of infection | Symptomatology | Symptoms duration (days) | DPSO | Reinfection before extraction day | Predicted variant | Vaccination status on extraction day |
|---|---|---|---|---|---|---|---|---|---|---|---|---|
| 1 | F | 35 | A*01, A*30, **B*08**, B*18 | Antigen test | N/A | Before vaccination | Cough, shortness of breath, anosmia | 10 | 90 | Yes | Delta | 1× mRNA-1273 |
| 2 | F | 27 | A*03, A*29, **B*08**, B*44 | Serology | N/A | Before vaccination | Cough, fatigue | Not reported | 440 | No | Wuhan-1 | 3× BNT162b2 |
| 3 | M | 22 | **A*02, A*24, B*08**, B*18 | PCR | 28.60 | Before vaccination | Fever, cough | 8 | 133 | No | Delta | Unvaccinated |
| 4 | M | 27 | A*01, **A*02, B*08**, B*50 | PCR | 20.32 | Before vaccination | Cough, night sweats, low-grade fever, lower back pain, fatigue | 7 | 448 | Yes | Alpha/Other | 1× mRNA-1273 |
| 5 | F | 32 | A*29, **A*68, B*07**, B*35 | PCR | 22.12 | Before vaccination | Cough, runny nose, fatigue | 5 | 182 | Yes | Alpha | Unvaccinated |
| 6 | M | 27 | A*30, **A*68**, B*41, B*55 | Antigen test | N/A | After 1st dose of vaccine | Fever, cough, anosmia, ageusia | 7 | 181 | No | Delta | 3× BNT162b2 |
| 7 | F | 23 | A*01, A*32, **B*08**, B*40 | PCR | 19.82 | After vaccination | Cough, runny nose | 4 | 24 | No | Omicron | 3× BNT162b2 |
| 8 | M | 26 | **A*02, A*24**, B*35, B*44 | PCR | 19.95 | After vaccination | Cough, runny nose | 5 | 40 | No | Omicron | 3× BNT162b2 |
| 9 | F | 27 | **A*02,** -, **B*07**, B*44 | Serology | N/A | Before vaccination | Cough | 2 | 650 | No | Wuhan-1 | 3× BNT162b2 |
| 10 | M | 25 | A*03, A*30, **B*08**, B*18 | PCR | 17.52 | Before vaccination | Fever, cough, headaches, fatigue | 8 | 140 | No | Delta | 1× BNT162b2 |
| 11 | F | 25 | A*01, A*11, **B*08**, B*55 | PCR | 16.06 | Before vaccination | Fever, headaches, odynophagia, myalgia | 6 | 310 | No | Alpha | 2× BNT162b2 |
| 12 | F | 27 | **A*02,** A*23, B*18, B*49 | PCR | 33.05 | Before vaccination | Headaches | 1 | 644 | No | Wuhan-1 | 3× BNT162b2 |

Bold values refer to the HLA-A, B used as inclusion criteria, mentioned in the Study Design section.

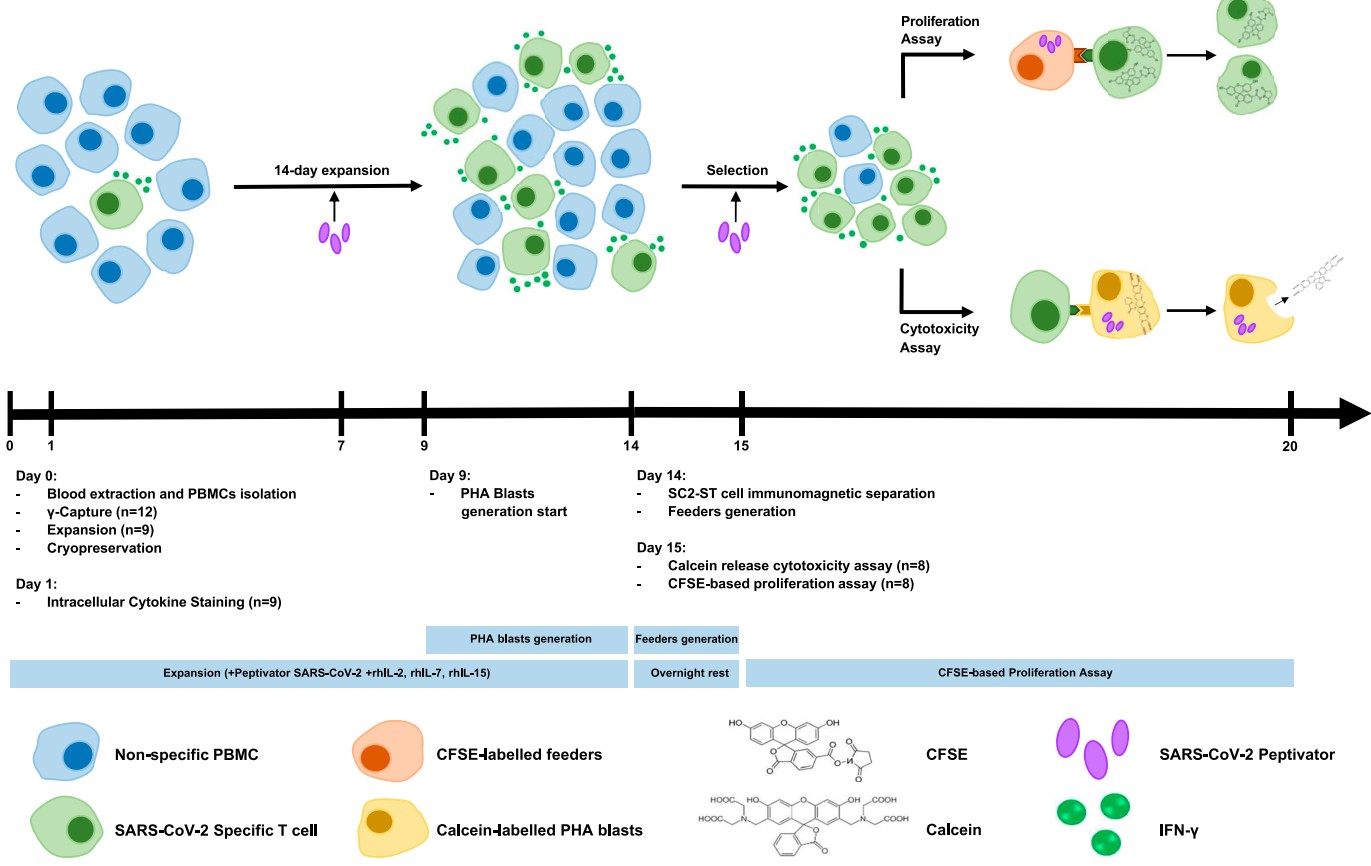

**Figure 1. Study Design.**
Cells were processed ex vivo for 3 wk. Starting at the day of peripheral blood extraction, PBMCs were stimulated with SARS-CoV-2 Peptivator to quantify SC2-STs in each donor, and the 14-d in vitro expansion of SC2-STs was performed with cell samples from subjects that met the criteria for clinical application of T lymphocyte donation. Remaining PBMCs were cryopreserved. On day 1, after overnight storage at 4°C, the phenotypical characterization of SC2-STs was performed. From day 9–15, PHA blasts from thawed PBMCs were generated. On day 14, the immunomagnetic enrichment of SC2-STs and the feeder generation from thawed PBMCs were performed. Isolated cells were maintained at 4°C overnight before using them in functional assays for the determination of their cytotoxic and proliferative capacities.

To characterize the memory differentiation status of SC2-STs, T cell subsets were described as: Naïve ($T_N$: CD45RO⁻CD45RA⁺CCR7⁺), central memory ($T_{CM}$: CD45RO⁺CD45RA⁻CCR7⁺), effector memory ($T_{EM}$: CD45RO⁺CD45RA⁻CCR7⁻), and late effector memory ($T_{EMRA}$: CD45RO⁻CD45RA⁺CCR7⁻). SC2-STs showed predominantly a $T_{EM}$ phenotype, with $T_{EMRA}$, $T_{CM}$, and $T_N$ cells representing ~1 10th of the specific population each (Fig 4C). The prevalence of the $T_{EM}$ phenotype was maintained in both CD4⁺IFN-γ⁺ and CD8⁺IFN-γ⁺ subsets. Comparatively, CD4⁺IFN-γ⁺ had a higher proportion of $T_{CM}$ phenotype, whereas CD8⁺IFN-γ⁺ exhibited an increase of both $T_{EMRA}$ and $T_N$ cells. No correlation was found between DPSO and the expression of these phenotypical markers, and no significant differences were observed between sexes (Figs S1 and S2). CD3⁺IFN-γ⁺Ki-67⁺ positively correlated with the proportion of specific $T_N$ cells ($P = 7.2 \times 10^{-3}$) and negatively with $T_{EM}$ cells ($P = 0.021$).

Together, these results indicate that SARS-CoV-2 convalescent donors can generate detectable IFN-γ in vitro after stimulation with the virus peptides. These antigen-specific cells mainly have an effector memory phenotype and basal expression of cytotoxic and activation markers.

## SC2-STs expand ex vivo upon antigen stimulation and can be enriched by MACS technology

Freshly isolated PBMCs were expanded for 14 d with recombinant human interleukin-2 (rhIL-2), interleukin-7 (rhIL-7), and interleukin-15 (rhIL-15) after stimulation with SARS-CoV-2 peptides (Fig 5A and B). There was a broad variation in the expansion rate of SC2-STs between donors, and it was neither influenced by sex nor DPSO (Figs S1 and S2). After the 14-d expansion, the percentage of SC2-STs increased from 0.0196% ± 0.0105% at day 0–3.92% ± 2.38% (Fig 5C). The negative control exhibited a percentage of SC2-STs of 0.0395% ± 0.0270%, showcasing the specificity of the expansion. Fold expansion of CD3⁺IFN-γ⁺ cells had a median of 294.4 ± 329.8. Similarly, the CD4⁺IFN-γ⁺ and CD8⁺IFN-γ⁺ subsets had median fold expansions of 389.2 ± 294.4 and 497.0 ± 741.1, respectively (Fig 5D). Great variability between the subjects was found in all three subsets, particularly in the CD8⁺IFN-γ⁺ population, which showed a CV of 94.2% compared with 71.3% for CD3⁺IFN-γ⁺ cells and 60.1% for CD4+IFN-γ⁺. At the end of the 14-d expansion, the CD4⁺IFN-γ⁺/CD8⁺IFN-γ⁺ ratio increased to 2.25 ± 2.57, showing great variability because of the generation of a highly CD4⁺IFN-γ⁺ enriched SC2-ST population in donors 3 and 5 (Fig 5E).

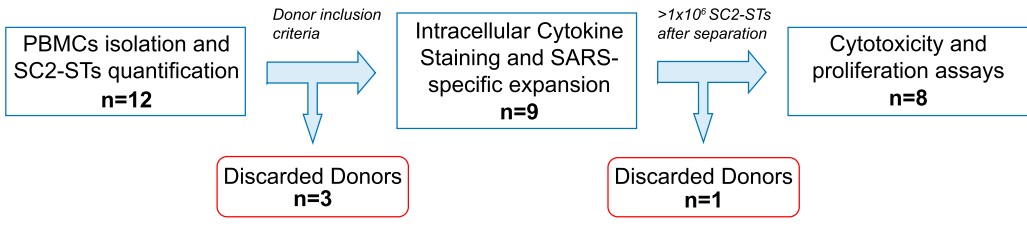

Figure 2. Changes in sample size throughout the study.
Donors were discarded on day 0 if some of the donor inclusion criteria were not met. Discarded donors were not further characterized. SC2-STs were then analyzed by ICS, expanded, and isolated through immunomagnetic separation. If the number of obtained specific cells was below $1 \times 10^6$ after isolation, the cytotoxicity and proliferation capacities were not analyzed.

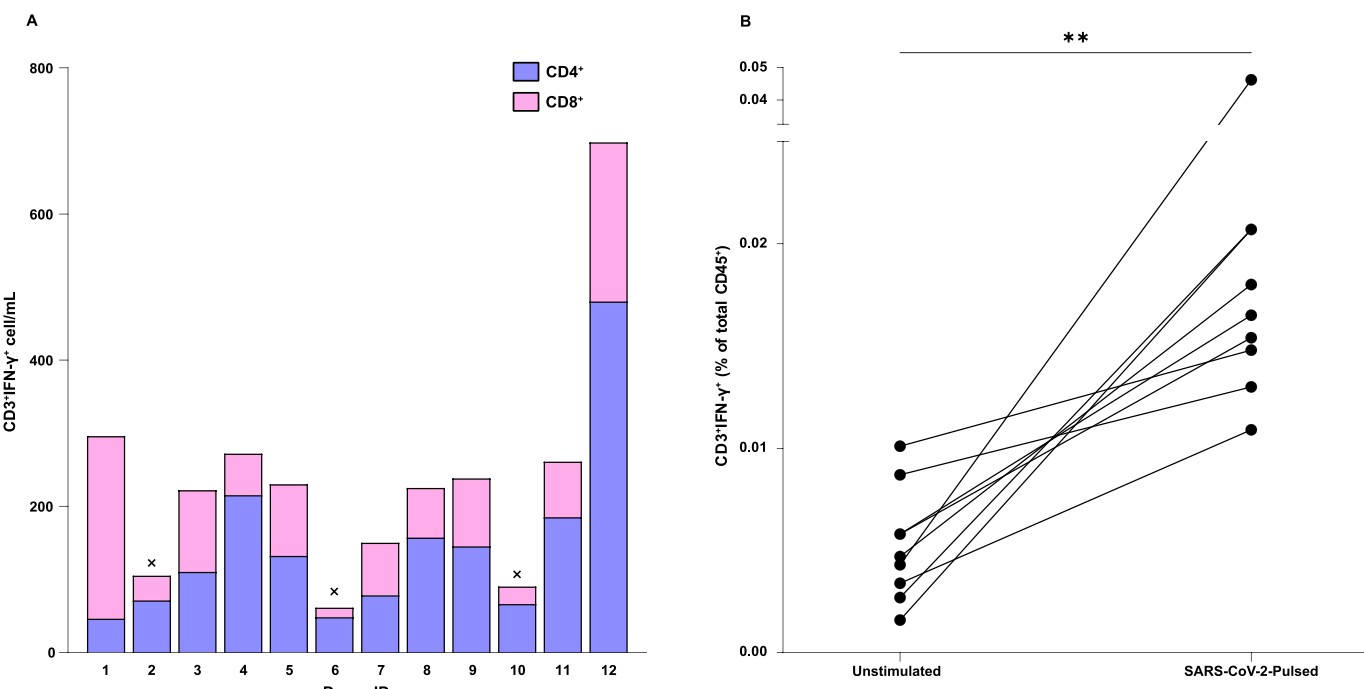

Figure 3. IFN-γ production by SC2-STs in convalescent donors.
(A) Absolute counts of IFN-γ-producing CD4+ and CD8+ cells in each convalescent donor. Donors 2, 6, and 10 did not meet the inclusion criteria and were excluded from further analysis. (B) Summary data for IFN-γ secretion in CD3+ T cells from SARS-CoV-2 convalescent donors ($n = 9$), unstimulated and stimulated with SARS-CoV-2 Peptivator. Both data sets were compared by paired parametric $t$ test **$P < 0.01$. Shown percentages come from the total viable CD45+ population.

The in vitro expansion was repeated with thawed cells from a small study group of four donors (donors 8, 9, 11, and 12) to phenotypically characterize the expanded SC2-STs on day 14 (Fig 2B). The vast majority of expanded SC2-STs presented an increase in $T_{EM}$ phenotype from day 0 ($P = 8.1 \times 10^{-4}$; 0.017; $<1 \times 10^{-4}$, for CD3+ IFN-γ+; CD4+ IFN-γ+, CD8+ IFN-γ+, respectively), with residual percentages from the rest of memory populations. These cells presented a significant up-regulation of activation (CD38, PD-1) and proliferation markers (Ki-67); and a reduction in granzyme B production. No significant differences were found in the percentage of perforin and FasL.

Next, the immunomagnetic enrichment of IFN-γ-secreting cells was carried out after SARS-CoV-2 peptide stimulation. Purified SC2-STs were obtained from all donors, with a purity of 75.50% ± 11.00% and yield of 79.23% ± 45.20% (Fig 5C). Yield was defined as the proportion of the absolute number of IFN-γ-producing cells present in the positive fraction and the absolute number of IFN-γ-producing cells in the sample before enrichment. When two consecutive enrichment processes were carried out, a decrease in the final yield was observed ($P = 0.046$), yet this led to a significant increase in the purity of the final product ($P = 5.3 \times 10^{-3}$) (Fig 5F and G). The overall CD4+IFN-γ+/CD8+IFN-γ+ ratio decreased after the enrichment process to 1.84 ± 2.17, with donors 3 and 5 maintaining their CD4+IFN-γ+ predominance (Fig 5E). Interestingly, these donors are the only subjects in the study group that were not vaccinated, so we performed a Fisher exact test to characterize this association. We found that the CD4+IFN-γ+/CD8+IFN-γ+ ratio is strongly associated with the vaccination status ($P < 0.001$), showing that unvaccinated subjects present a higher CD4+IFN-γ+/CD8+IFN-γ+ ratio. In donor 4, the expansion and selection led to an insufficient number of SC2-STs to use in functional assays, so it was not functionally characterized (Fig 2).

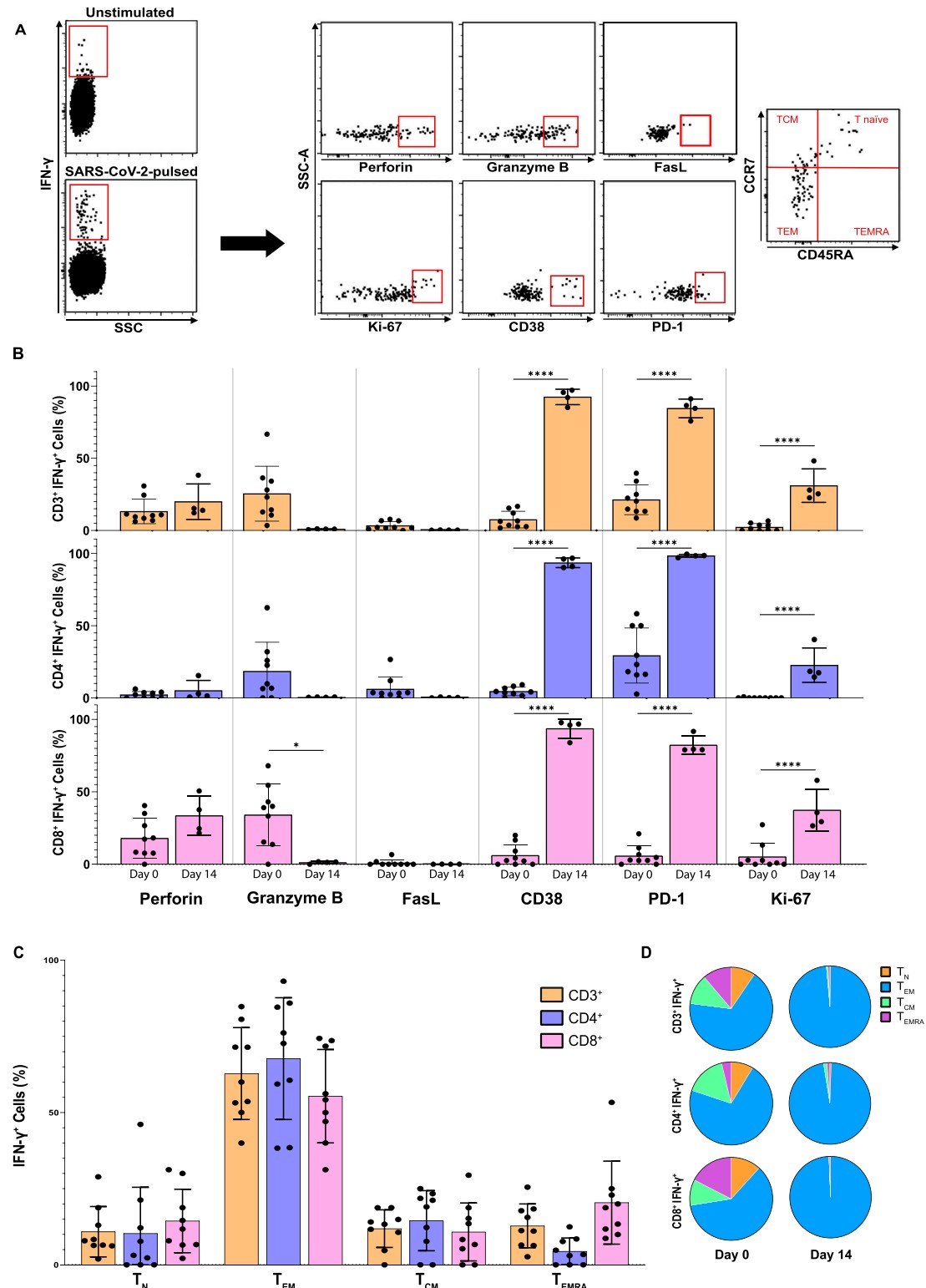

**Figure 4. Phenotypical characterization of SC2-STs.**
**(A)** Representative flow cytometry plots for cells expressing IFN-γ after SARS-CoV-2 peptide stimulation, gated on viable CD3$^+$CD45$^+$ cells, CD3$^+$CD4$^+$, and CD3$^+$CD8$^+$.
**(B)** Expression percentage of multiple markers in SARS-CoV-2-specific IFN-γ-producing CD3$^+$, CD4$^+$, and CD8$^+$ subsets, obtained by intracellular cytokine staining from SARS-CoV-2 convalescent donors at the blood extraction day (n = 9) and after the 14-d expansion (n = 4), compared by paired parametric t test *P < 0.05,****P < 0.0001.
**(C)** Memory phenotype of SARS-CoV-2-specific IFN-γ-producing CD4$^+$ and CD8$^+$ cells at day 0, including T Naïve (T$_N$; CD45RO$^-$CD45RA$^+$CCR7$^+$), T Central Memory (T$_{CM}$; CD45RO$^+$CD45RA$^-$CCR7$^+$), T Effector Memory (T$_{EM}$; CD45RO$^+$CD45RA$^-$CCR7$^-$), and T Late Effector Memory (T$_{EMRA}$; CD45RO$^-$CD45RA$^+$CCR7$^-$). **(D)** Proportion of T cell memory subpopulations in SC2-STs at day 0 and at day 14 after expansion.

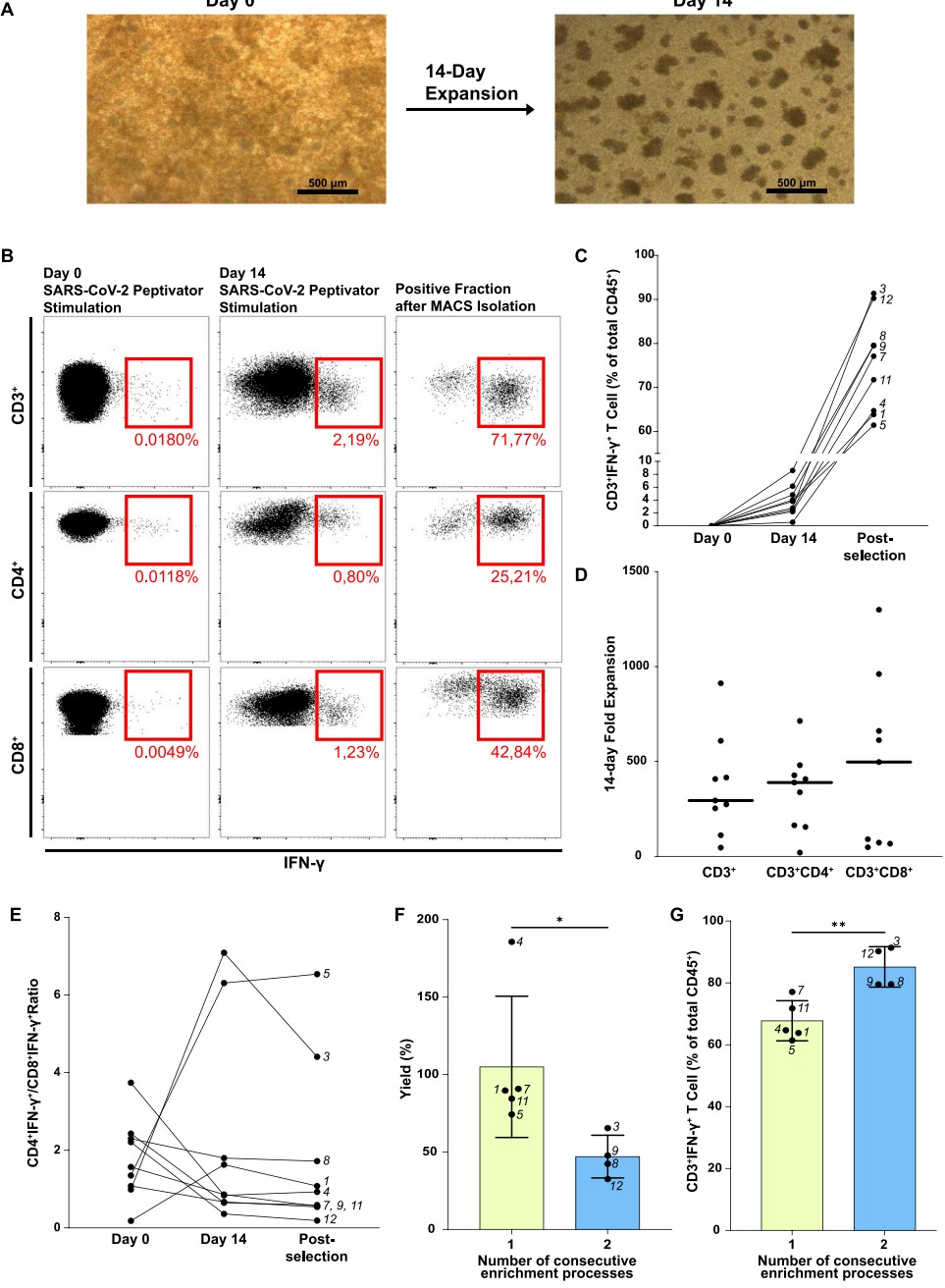

**Figure 5. Expansion and enrichment of SC2-STs in vitro.**
**(A)** Representative microscopy images (10X) of in vitro expansion of PMBCs at day 0 and at day 14. **(B)** Representative flow cytometry plots of the percentage of SARS-CoV-2-specific IFN-γ$^+$ cells in the CD3$^+$, CD4$^+$, and CD8$^+$ subsets on day 0, day 14 after expansion and after immunomagnetic separation. Gates originate from viable CD45$^+$CD14$^-$CD20$^-$CD3$^+$ cells, and shown percentages come from the overall CD45$^+$ population. **(C)** Summary data of the increase of IFN-γ expression in CD3$^+$ cells at different time points during sample processing. **(D)** Fold expansion of SARS-CoV-2 specific CD3$^+$, CD4$^+$, and CD8$^+$ T cells. **(E)** Changes in the specific CD4$^+$IFN-γ$^+$/CD8$^+$IFN-γ$^+$ ratio during sample processing (n = 9). **(F, G)** Yield and purity of the immunomagnetic separation process when expanded cells are isolated through one or two consecutive enrichment processes.

Using the four-donor study group, the selected product was tested for the presence of NK cells. Because of a complete CD3$^+$ internalization caused by the expansion and activation associated to immunomagnetic enrichment, NK and NKT populations could not be differentiable The obtained SC2-ST percentage of the enriched product in this study group was 63.4% ± 11.9%. We found a median of 18.13% ± 8.53% CD56$^+$ cells in the remaining IFN-γ- population. To discern the nature of this CD56$^+$ population, we also analyzed said marker before SC2-ST isolation (in which, CD3$^+$ internalization did not hinder the analysis). We observed that there were eight times more NK cells than NKT cells in the IFN-γ- population.

Therefore, SC2-STs have the potential to expand in vitro, and their immunomagnetic enrichment through IFN-γ capture methodology generated a specific and enriched T cell population.

### Enriched SC2-STs show antigen-specific cytotoxicity in vitro, dependent on the target to effector ratio and on specific CD8$^+$ proportion

Next, we assessed the cytotoxic potential of isolated effector SC2-STs through a Calcein-release-assay with autologous SARS-CoV-2-

pulsed PHA blasts as target cells and cocultured at different target to effector ratios. Enriched cells successfully lysed pulsed PHA blasts, showing SARS-CoV-2-specific cytolytic activity dependent on the number of SC2-STs (Fig 6A). When cocultured with unstimulated PHA blasts, a significant reduction in Calcein release was observed ($P$ = 6.0 × 10$^{-3}$; 0.017; 0.046, for 1:10, 1:5 and 1:1 ratios, respectively). Similarly, a reduction in Calcein release was observed when SARS-CoV-2-pulsed PHA blasts were previously blocked with anti-HLA class I and II antibodies, but this difference did not reach significance in any of the studied ratios.

Cytotoxicity is a complex process where multiple factors can confound the outcome. To characterize the specific effect of the treatment (target cell stimulation) over cytotoxicity, we adjusted a generalized mixed multivariate model accounting for the variables that had a significant effect over cytotoxicity between the following potential confounders: CD4$^+$IFN-γ$^+$/CD8$^+$IFN-γ$^+$, target cell/effector cell ratio, age of the donor, sex, DPSO, and CD3$^+$IFN-γ$^+$ percentage after enrichment. Once we established the model, we set the SARS stimulated treatment as the reference treatment, allowing us to measure change in cytotoxicity when target cells are not stimulated (negative control), and when HLA is blocked. We found a significant decrease in the specific lysis after HLA blockade by 8.9 units (CI$_{95\%}$ [−13.41; −4.54]; $P$ < 0.001), whereas the cytotoxicity in the negative blast control decreased by 16.6 units (CI$_{95\%}$ [−21.14; −12.15]; $P$ < 0.001). Similarly, we found a significant positive effect of the target to effector ratio of 1.21 (CI$_{95\%}$ [0.71; 1.70]; $P$ < 0.001). No significant effect was found caused by age, sex, DPSO, and CD3$^+$IFN-γ$^+$ after enrichment.

When analyzed individually, donors 3 and 5 showed much lower cytotoxic potential, with marginal difference between SARS-CoV-2-pulsed blasts and unstimulated blasts. As mentioned before, these donors had larger quantities of specific CD4$^+$ cells compared with specific CD8$^+$. We observed that there was a negative correlation ($P$ = 0.021) between the cytotoxic potential of SC2-STs against SARS-CoV-2-pulsed blasts at 1:10 and its specific CD4$^+$ IFN-γ$^+$/CD8$^+$ IFN-γ$^+$ ratio (Fig 6B). This circumstance was also found in the multivariate analysis, showing a negative effect in the specific cytotoxicity of the CD4$^+$ IFN-γ$^+$/CD8$^+$ IFN-γ$^+$ ratio of -2.93 (CI$_{95\%}$ [−5.19; −0.67]; $P$ < 0.012).

These findings demonstrate that SC2-STs show antigen-specific cytotoxicity in vitro, which depend on the specific CD8$^+$ proportion of the final enriched population and the target to effector ratio in culture.

### Enriched SC2-STs are capable of proliferation in vitro after antigenic rechallenge and retain SARS-CoV-2-specificity

To determine the proliferative response of SC2-STs after an antigenic rechallenge, a CFSE dilution assay was performed, coculturing the enriched cells with SARS-CoV-2 peptide-pulsed feeders for 5 d (Fig 6C). SC2-STs showed an important reduction of CFSE's median fluorescence intensity at the unpulsed control in all donors and subpopulations from day 1 to 5 (Fig 6C), showing that the isolation process activates the specific subpopulation. After performing a paired statistical analysis of these data, an additional significant median fluorescence intensity reduction ($P$ = 3.6 × 10$^{-3}$) in the CD3$^+$ population was observed between the unpulsed control and the SARS-CoV-2 peptide-pulsed condition (Fig 6D) despite the

heterogeneity between samples. Similarly, specific CD8$^+$ cells showed a significant reduction ($P$ = 6.2 × 10$^{-3}$) in all subjects, whereas CD4$^+$ cells showed this reduction in 7/8 subjects but did not reach significance ($P$ = 0.17).

Thus, enriched SC2-STs maintained their proliferation potential upon antigen presentation in spite of the activated state induced by the isolation methodology.

## Discussion

The importance of antigen-specific T cells in virus clearance after acute infection is widely demonstrated in human studies, with both CD4$^+$ and CD8$^+$ subpopulations playing an essential role (Schmidt & Varga, 2018; Kervevan & Chakrabarti, 2021). Considering severe SARS-CoV-2 infection, in which patients show lymphopenia and T-cell exhaustion (Xu et al, 2020), allogenic cellular therapy with SC2-STs arises as an attractive alternative for the treatment of hospitalized severe cases and their immune reconstitution. Because of their biological complexity, an exhaustive characterization of these candidate cells should be performed before clinical application to predict their inherent bioactivity, including their phenotype and functional capacities.

Consistent with previous studies (Sekine et al, 2020; Breton et al, 2021), here, we have demonstrated that SC2-STs from convalescent subjects produce IFN-γ upon in vitro antigenic challenge, and 75% of recruited individuals showed sufficient CD3$^+$IFN-γ$^+$ to carry out a functional cell characterization. As described in other studies (Cooper et al, 2021; Yeo et al, 2021), the IFN-γ-producing population is predominantly CD4$^+$. We used a peptide pool covering the whole proteome of the Wuhan WT SARS-CoV-2 virus, generating a polyclonal immune response minimally affected by antigenic escape, thus being effective against future VOCs. SC2-STs were detected in donors in which their primary infection was most likely caused by different variants, including the ancestral virus and the alpha, delta, and omicron variants. In addition, we were able to generate SC2-STs from donors with different HLA, including HLA-A*02, A*24, A*68, B*07 or B*08. Lastly, it is also worth noting that no significant differences were found in IFN-γ production or any other experimental condition between females and males.

Our study group is composed of young donors (range 22–35 yr) with mild symptomatology and a range of 24–650 d elapsed from infection to the day of extraction. Prior studies showed that the presence and functionality of SC2-STs are retained up to 1 yr after primary acute infection (Dan et al, 2021; Guo et al, 2022). Indeed, we did not find a correlation between DPSO and IFN-γ production by SC2-STs at blood extraction day. In addition, we could obtain SC2-STs cells from donors up to 650 d postinfection, demonstrating long-lasting cellular immunity in COVID-19 young convalescent donors. Multiple studies have demonstrated that subjects with higher age ranges (Peluso et al, 2021) and donors with asymptomatic primary infection (Sekine et al, 2020) can also generate T cell memory responses with SARS-CoV-2-specific IFN-γ production. Lastly, it is worth asking whether uninfected individuals would be eligible for SC2-ST donation, because of the presence of conserved epitopes in the spike protein and crossed reactivity with

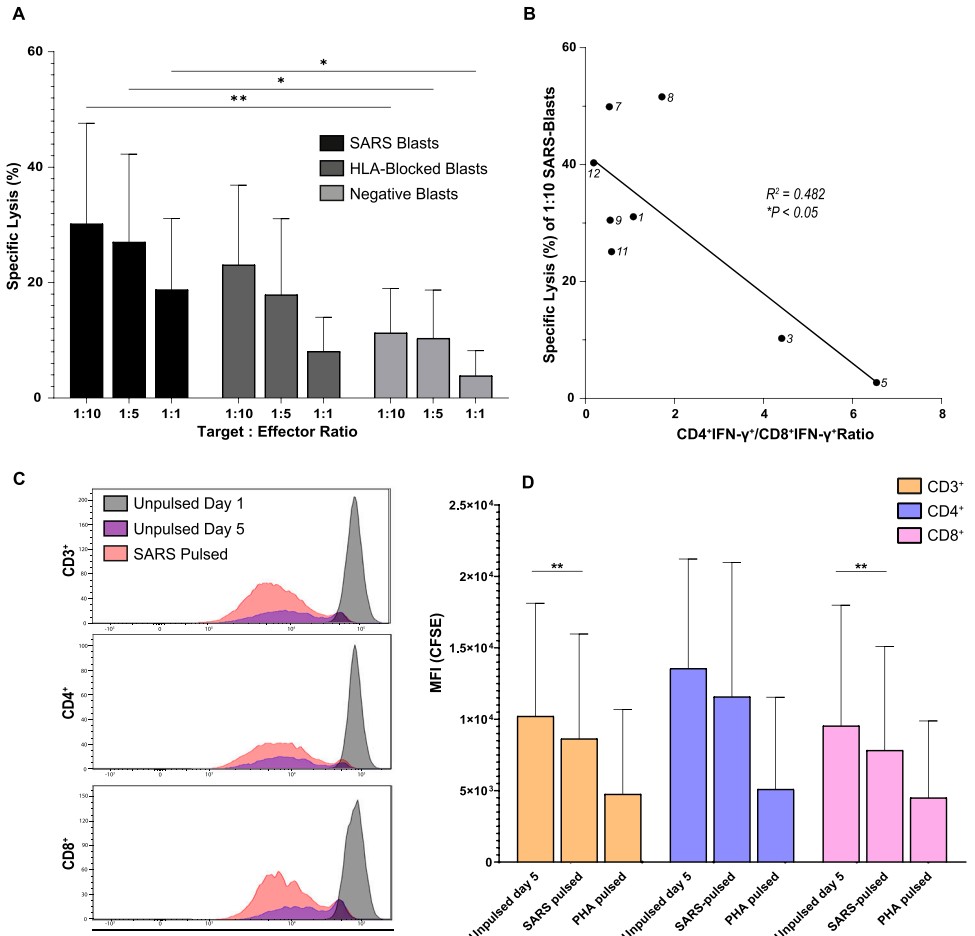

**Figure 6. Functional characterization of enriched SC2-STs.**
**(A)** Summary data of the calcein-release cytotoxicity assay (n = 8), showing the specific lysis induced by SC2-STs when cocultured with target PHA blasts at different target to effector ratios. SARS-CoV-2 peptide-pulsed PHA blasts (SARS Blasts), SARS-CoV-2 peptide-pulsed PHA blasts blocked with anti-HLA class I and anti-HLA class II antibodies (HLA-Blocked Blasts), and unstimulated PHA blasts (Negative Blasts) were used for each experimental condition. An ordinary two-way ANOVA with Tukey's multiple comparison test was performed *P < 0.05; **P < 0.01. **(B)** Correlation analysis between the specific lysis of SARS Blasts at 1:10 target to effector ratio and the SARS-CoV-2-specific CD4⁺/CD8⁺ ratio after immunomagnetic separation, using a parametric Pearson correlation test. **(C)** Representative flow cytometry histograms of the CFSE-based assay (n = 8), showing the CFSE dilution in isolated CD3⁺, CD4⁺, and CD8⁺ SC2-STs during the 5-d coculture with SARS-CoV-2 peptide-pulsed (SARS Pulsed) and unpulsed autologous feeders. **(D)** Summary data of the proliferation assay, showing the MFI in the negative control, the SARS-pulsed condition and the PHA-pulsed positive control in the CD4⁺, CD8⁺, and overall CD3⁺ populations. Both experimental conditions were compared in each subset by paired parametric t test *P < 0.05.

common cold coronaviruses. It has been demonstrated that there is a T memory repertoire against a spike in uninfected donors (Grifoni et al, 2020b; Ogbe et al, 2021), but these cells have reduced functionality and cytotoxic potential (Kim et al, 2021), and it is unlikely that these donors would generate an adequate number of SC2-STs to meet the T-lymphocyte donation criteria without prior antigen-specific expansion.

When analyzed by intracellular staining, IFN-γ-producing SC2-STs showed mainly $T_{EM}$ phenotype, followed by $T_{EMRA}$ in the CD8⁺ subset, and by $T_{CM}$ in CD4⁺ cells, as established by other studies with SARS-CoV-2 convalescent and vaccinated donors (Guerrera et al, 2021; Kared et al, 2021). The immediate effector function and tissue-homing capacity of $T_{EM}$ cells grant them an essential role in the immune reaction against an acute infection, so the prevalence of this memory subpopulation in the final T cell therapy product represents an important asset when fighting severe SARS-CoV-2 infection. In addition, the presence of cell memory subpopulations like CD4⁺ $T_{CM}$ and CD8⁺ $T_{EMRA}$ may be beneficial in a hypothetical GMP-compliant adoptive therapy product, combining the great expansion potential in vivo of the former and the high cytotoxic function of the latter. Although a small proportion of naïve cells were also found in both subsets, an advantage of the IFN-γ capture methodology over other VST-isolation approaches is that the

obtained cellular product is composed of highly specific memory cells, with residual potentially alloreactive naïve cells. This implies that it will be unlikely to generate an alloreactive response after clinical T cell infusion, even with a small proportion of naïve cells. Lastly, similar to what was previously reported (Basar et al, 2021), the in vitro expansion generated a strong $T_{EM}$ population. It should be noted that if this $T_{EM}$ expansion is carried out in vivo, a hypothetical cell therapy product based on these cells should be administered in multiple doses, compared with a $T_{CM}$-based product. Although many more cellular microenvironment factors play a role in the in vivo expansion, we have shown the capacity of SC2-STs to acquire in vitro an effector phenotype after a new SARS-CoV-2 antigenic stimulus.

As expected, SC2-ST CD8⁺ cells exhibited higher expression of perforin and granzyme B than their CD4⁺ counterpart. This reduction is particularly important in CD4⁺IFN-γ⁺perforin⁺ cells, which displayed an 8.3 times lower expression than CD8⁺-specific cells. This phenomenon is probably justified by the loss of the perforin–granzyme B association in CD4⁺ memory cells, as described in a previous study (Lin et al, 2014). Obtained marker frequencies are also analogous to those reported in previous studies (Chen et al, 2021; Liu et al, 2021), that concluded that these cytotoxic molecules are up-regulated in the general CD4⁺ and CD8⁺

subpopulations on acute SARS-CoV-2 infection and decrease in the convalescent phase of the infection, reaching similar values to those obtained in uninfected subjects. The percentage of FasL$^+$ cells in SC2-STs is substantially smaller than granzyme B and perforin, and it is mostly undetectable in specific CD8$^+$ cells. The role of FasL in SARS-CoV-2 infection has not been extensively studied. However, a recent study (André et al, 2022) reported an increase of soluble FasL in plasma and of FasL membrane expression during acute infection, and linked it to COVID-19-associated lymphopenia. The return to normal levels of T-cells in the convalescent phase of the infection may be associated with this limited expression of FasL. The percentage of these cytotoxic markers was not significantly affected by the expansion, with the exception of granzyme B, which was reduced despite an increase in the number of effector cells. The absence of type I IFNs and IL-12 in the expansion protocol, which favor the expansion of cytolytic phenotype (Curtsinger et al, 2005; Nowacki et al, 2007), could explain this circumstance. In spite of this decrease, it was possible to observe that SC2-STs continue to exert antigen-specific cytotoxicity in subsequent cytotoxicity assays.

SC2-STs also exhibited limited expression of CD38 in convalescent donors. It has been demonstrated that this enzyme is up-regulated in hyper-inflammation states, playing an essential role in immune responses, including cellular migration and antigenic presentation (Piedra-Quintero et al, 2020). Indeed, CD38 is highly up-regulated in T lymphocytes of COVID-19 acute patients (Sekine et al, 2020), and decreases over time once the infection is resolved and the inflammation state disappears (Rha et al, 2021). Here, PBMCs from most convalescent donors were obtained at least 3 mo after disease onset, and SC2-STs from said donors may have reached the basal expression of this marker, with no ongoing activation/inflammation state. Similarly, Ki-67 and PD-1 are also up-regulated in T lymphocytes from SARS-CoV-2 infected subjects, especially in severe cases (Sekine et al, 2020), and their expression is reduced once the infection is resolved (Breton et al, 2021; Rha et al, 2021). SC2-STs showed low or undetectable number of CD4$^+$Ki-67$^+$, CD8$^+$Ki67$^+$, and CD8$^+$PD-1$^+$ cells, demonstrating again that these cells may have returned to their preinfection state, with the notable exception of PD-1 expression in SC2-ST CD4$^+$ cells, which exhibited greater percentage than most of the analyzed markers. This phenomenon has previously been reported in latent CMV donors (Parry et al, 2021), which showed high percentages of CD4$^+$PD-1$^+$ cells that remained stable over time, had great variability between donors and correlated with the viral load in acute infection. This does not apply to CD8$^+$ cells, in which, the CD8$^+$PD-1$^+$ percentage decreases with DPSO, similarly to CD38. As expected, after antigen-specific expansion, SC2-STs show an activated phenotype with proliferative capacity, acknowledging the capacity of these cells to activate and proliferate after an antigenic rechallenge in vitro. An elevated expression of PD-1 in T lymphocytes can also be associated with a senescent phenotype (Janelle et al, 2021), but further studies should be carried out to evaluate this possibility.

In summary, SC2-STs exhibited basal frequencies of most of the cytotoxic and activation markers, possibly reached after an up-regulation of said markers during acute COVID-19 as described in numerous studies in the general T lymphocyte memory population. Thus, these specific cells do not exhibit a functional phenotype different from that found in the overall functional T lymphocyte population. The prevalence of $T_{EM}$ and the modest expression of cytotoxicity and activation markers led to no significant correlations between said markers and memory subsets, with the exception of CD3$^+$IFN-γ$^+$Ki-67$^+$, that positively correlated with $T_N$ cells and negatively with $T_{EM}$ cells, emphasizing the differences in the proliferation capacities of said memory phenotypes (Redeker & Arens, 2016).

The in vitro expansion of SC2-STs was carried out successfully, obtaining analogous results to those reported in prior studies (Keller et al, 2020; Guerreiro et al, 2021) and obtaining a sufficient number of specific cells for their functional analysis in 8/9 donors. The CD4$^+$ and CD8$^+$ composition of the expanded product was heterogeneous because of great variability in the expansion of the CD8$^+$ subset, a circumstance that has been observed in other viruses like CMV (Grau-Vorster et al, 2020). We have demonstrated that the immunomagnetic isolation of specific T cells based on IFN-γ is applicable to SC2-STs, showing a great performance and obtaining a highly enriched and valuable cell population.

SC2-STs showed specific cytotoxicity when cocultured with autologous peptide-pulsed PHA blasts, and a significant reduction was observed when unpulsed PHA blasts were used. The observed cytotoxicity in the negative experimental condition was low, demonstrating that the cytotoxic activity of isolated cells is highly specific to SARS-CoV-2 specific peptides. The in vitro cytotoxic potential of the isolated cells was also highly dependent of its CD4$^+$/CD8$^+$ ratio, as theorized in a previous study with SC2-STs (Kim et al, 2021), demonstrating here that cell products with a CD4$^+$ dominance performed poorly in the cytotoxicity assays. In addition, we have shown that this CD4$^+$ dominance is associated with the vaccination status of subjects, demonstrating the increasing importance of incorporating the immune status (natural immunity, vaccine-induced immunity, hybrid immunity) in immunological studies. Future studies are necessary to characterize the effect of this immunity, not only in cytotoxicity but also in additional aspects of the cellular immunity.

As previously mentioned, the HLA plays an essential role in the SARS-CoV-2 specific immunological reaction. Here, we have found a significant negative association between the specific cytotoxicity and the HLA blockage of the target cells, analyzed by multivariate analysis. This decrease in the cytotoxicity does not match the effect of the negative blasts, indicating that HLA-independent cytotoxicity is likely taking place. NK cells are an important component of innate immunity, with HLA-independent cytotoxic demonstrated capacity (Quatrini et al, 2021), so we analyzed their presence in the effector population. Indeed, we found a small but relevant population of NK/NKT cells among the non-SC2-ST population which may be responsible for this HLA-independent cytotoxicity. The previously mentioned influence of CD4$^+$ cells on cytotoxicity may have affected these results as well. Despite the presence of this HLA-independent cytotoxicity, the specific cytotoxicity exerted by SC2-STs remained predominant, as demonstrated by the multivariate analysis. Thereby, we have demonstrated the antigen-specific cytotoxic potential of SC2-STs, which may strengthen the potency of the cellular product in a clinical setting.

The enriched product also maintained the proliferative capacity of SC2-STs, previously showed in the expansion, as demonstrated in

the CFSE-based proliferation assay. Because the isolation method used in this study relies on the functional activation of specific cells for IFN-γ secretion, SC2-STs are in a preactivated state and exhibited proliferative responses when cocultured with unpulsed feeders. However, an additional stimulation in vitro with SARS-CoV-2-pulsed feeders significantly increased the proliferation of SC2-STs, showing its antigen-specific effect. In a clinical setting, a different VST isolation approach that does not rely on the isolation of functionally active cells, like HLA-I multimer technologies, would probably generate a limited repertoire of heterogeneous, unstimulated specific CD8$^+$ cells. However, the significant presence of the CD4$^+$ subset may be essential to generate and sustain the CD8$^+$ antiviral response in vivo after adoptive T cell infusion (Novy et al, 2007; Laidlaw et al, 2016; Ahrends et al, 2017), and it has been demonstrated that the infusion of IFN-γ-secreting preactivated T cells is a viable therapy for other virus like CMV and EBV (Moosmann et al, 2010; Peggs et al, 2011). However, it is yet to be established if this preactivated state denotes a clinical advantage over different VST isolation alternatives.

This investigation was conceptualized as a preclinical study, and it presented some limitations. First, the sample size, age, and symptomatology ranges of donors were limited. Our study group allowed the validation of SC2-STs as a T cell therapy candidate, but a larger, more diverse group should be used in future studies if a more in-depth characterization of these IFN-γ producing cells is desired. The use of SC2-STs obtained from uninfected donors (Swadling et al, 2022) was also not considered in this study, and further studies are needed to characterize SC2-STs obtained from this type of donor. Likewise, the possible presence of asymptomatic infection at the time of blood extraction was not documented. Furthermore, additional specific cell subpopulations found in SARS-CoV-2 convalescent and vaccinated individuals like Tregs and cTfh (Notarbartolo et al, 2021; Yeo et al, 2021; Samanovic et al, 2022) were not analyzed, because they will represent a small fraction in the T-cell therapy product, being unlikely that they will affect its therapeutic efficacy in a meaningful way. Further studies may be of interest to describe the relationship of said cellular subpopulations and SARS-CoV-2 immune responses. Lastly, some sensibility was lost in the determination of the SC2-STs phenotype, as consequence of the low number of detected IFN-γ-producing cells at peripheral blood extraction day.

SC2-ST percentages in convalescent subjects are two to four times smaller than other VSTs like influenza or CMV (Rha et al, 2021), the latter being one of the most important viruses in adoptive therapy studies. Despite this reduction, an important proportion of convalescent donors showed CD3$^+$IFN-γ$^+$ > 0.01%, the established threshold of virus-specific T-cells for clinical application. Indeed, a previous study (Leung et al, 2020) have demonstrated that clinical-grade SC2-STs can be obtained from convalescent donors using the IFN-γ capture technology, with a sufficient cell count to be used as T cell therapy in COVID-19 hospitalized patients. In the present study, we have additionally characterized the potency of these SC2-STs, which showed an effector memory phenotype associated to a basal expression of cytotoxicity, activation, and proliferation markers; and are functionally immunocompetent proved their capacity to expand in vitro, their specific cytotoxic function and their proliferative potential upon antigenic re-challenge. Proven their functionality and potency, SC2-STs could be used in the future for the manufacture of a GMP-compliant adoptive therapy against COVID-19-associated pneumonia.

# Materials and Methods

### Study design

The objective of this study was to perform a comprehensive characterization of functionally active SARS-CoV-2 specific T cells obtained from COVID-19 convalescent donors, as a preclinical study aimed to validate the use of these cells as an allogenic T cell therapy for the treatment of patients with severe SARS-CoV-2-associated pneumonia. Adult donors recovered from mild SARS-CoV-2 infection (n = 12) were recruited for blood donation. The study (PI_2020/68) was approved by the Ethics Committee for Drug Research of Navarra (CEIm). Informed written consents were obtained from all donors, following the guidelines of the local ethics committee and in accordance with the Declaration of Helsinki. All enrolled donors were previously diagnosed of SARS-CoV-2 infection by PCR in nasopharyngeal samples, positive IgM/IgG serology result or a positive antigen rapid test result; and did not show COVID-19-associated symptomatology on blood extraction day. The virus variants that most probably infected the subjects were classified according to the diagnostic date and epidemiological data obtained from the Spanish Health Ministry (Centro de Coordinación y Emergencias Sanitarias, 2022).

In addition, donor recruitment was limited to subjects with HLA-A*02, A*24, A*68, B*07 or B*08 antigens, proven their ability to present SARS-CoV-2 immunodominant epitopes and their presence in 50% of the Spanish population according to current literature (Montero-Martín et al, 2019; Grifoni et al, 2020a). Donor characteristics are listed in Table 1. In addition, the suitability of the donor cells was evaluated following three criteria for clinical application of virus-specific T cell therapy provided by the manufacturer of the IFN-γ detection kit. First, IFN-γ-producing CD3$^+$CD45$^+$ cell percentage must be superior to 0.01%; then, there must be at least 10 IFN-γ-producing CD3$^+$ cells for every $1 \times 10^5$ CD45$^+$ cells; finally, the CD3$^+$IFN-γ$^+$ cell count after peptide stimulation must double the number of the unstimulated control.

### Peripheral blood mononuclear cells isolation

150 ml of peripheral blood were extracted from each donor in blood collection bags with a citrate–phosphate–dextrose solution with adenine (CPDA-1) as the anticoagulant solution. PBMCs were isolated by density gradient centrifugation using Ficoll-Paque PLUS (GE Healthcare) and used within a day in the following assays. Remaining PBMCs were cryopreserved with FBS (Gibco) and 10% (vol/vol) DMSO (Thermo Fisher Scientific) and stored in liquid nitrogen at −196°C for later use in post-expansion phenotypical characterization assays and in proliferation and cytotoxicity functional assays. When needed, PBMCs were rapidly thawed in a 37°C water bath, centrifuged, and cultured 24 h before its use.

## HLA class I typing (HLA-A, -B)

DNA was extracted from peripheral blood samples using the Maxwell RSC Blood kit (Promega) following the manufacturer's recommendations. Loci A and B of HLA class I were genotyped by PCR-SSO (PCR-Sequence Specific Oligonucleotide) methodology using Luminex technology (Luminex Corporation) and the Lifecodes HLA-A and HLA-B SSO typing kits (Immucor GTI Diagnostics, Inc.).

Briefly, a PCR mixture was prepared using the Lifecodes Master Mix, the genomic DNA, and Taq polymerase. Then, the PCR was carried out with an initial denaturation step, 40 cycles of amplification and one cycle of extension. Next, hybridization was performed, adding 7.5 $\mu$l of probe mix to 2.5 $\mu$l of locus-specific PCR products. Finally, hybridized samples were diluted with streptavidin dilution buffer and analyzed within 30 min using the Luminex 200 System (Luminex Corporation). The probe-hit pattern was compared with the common and well-documented HLA alleles (IMGT-HLA Database 3.43) by using the MatchIT DNA program (Immucor GTI Diagnostics, Inc.).

## Quantification of SC2-STs based on IFN-γ secretion technology

$2 \times 10^6$ PBMCs were cultured in flat-bottom 96-well culture microplates (Corning) with RPMI 1640 medium (Gibco), 5% (vol/vol) AB serum (Sigma-Aldrich), and 1% (vol/vol) penicillin/streptomycin (Gibco). Cells were stimulated with 63 MHC class I restricted and 25 MHC class II restricted SARS-CoV-2 peptides (Peptivator SARS-CoV-2 Select; Miltenyi Biotec), originated from structural (Spike [S], Membrane [M], Nucleocapsid [N], Envelope [E]) and non-structural proteins, to a final concentration of 1 $\mu$g/ml. Peptivator SARS-CoV-2 Select includes HLA-A*02-, A*24-, A*68-, B*07-, and B*08-restricted peptides among others. Negative (unstimulated cells) and positive (stimulated cells with 40 $\mu$g/ml of *Staphylococcus aureus* enterotoxin B [SEB; Sigma-Aldrich]) controls were included.

After 4 h of incubation at 37°C, 5% (vol/vol) $CO_2$, cells were labelled with IFN-γ Catch Reagent (IFN-γ Secretion Assay - Detection Kit; Miltenyi Biotec), containing bispecific antibodies for human CD45 leukocyte marker and IFN-γ, and were incubated for 45 min to allow the secretion of IFN-γ by SC2-STs. Then, the cytokine secretion reaction was stopped adding ice-cold buffer, containing PBS (Gibco), 0.5% (vol/vol) BSA (Gibco), and 2 mM EDTA (Sigma-Aldrich). According to the manufacturer's instructions, the cells were labelled with a monoclonal human IFN-γ PE antibody for 10 min at 4°C in the dark.

Next, cell surface staining was performed using the following fluorochrome-conjugated antibodies: CD3-Violet Fluor 450, CD4-APC, CD8-PE-Vio770, CD14-VioGreen, CD20-VioGreen, CD45-FITC, and 7-Aminoactinomycin D (7-AAD) viability dye. Additional information on used antibodies is shown in Table S1. Before acquisition, residual red cells were eliminated with ACK lysing buffer (Thermo Fisher Scientific). Sample acquisition was performed on a three-laser FACS Canto II cytometer (Becton Dickinson). Gating strategy is shown in Fig S3.

## Phenotypic and functional characterization of SC2-STs

$2 \times 10^6$ PBMCs were resuspended in three round-bottom tubes (Thermo Fisher Scientific) with RPMI 1640 media, 0.1% (vol/vol), BSA (Sigma-Aldrich) and stimulated with Peptivator SARS-CoV-2 Select (1 $\mu$g/ml). Each tube was associated with a different multicolor flow cytometry panel. Negative and positive controls were also included. After 1 h of incubation at 37°C, 5% (vol/vol) $CO_2$, 10 $\mu$g/ml brefeldin A (Sigma-Aldrich), 2.5 $\mu$g/ml monensin (BioLegend), and fresh RPMI 1640 media supplemented with 12.5% (vol/vol) FBS were added and cultured for 4 h. Cell pellet was incubated with 0.02% (vol/vol) EDTA and then stained with fluorochrome-conjugated antibodies for cell surface labeling. The cells were then fixed and permeabilized with IntraStain kit (DAKO Agilent), followed by intracellular staining with human monoclonal antibodies. The antibodies used for cellular labeling were: CD3-Violet Fluor 450, CD3-VioGreen, CD3-VioBright 515, CD4-PerCP-Vio700, CD8-PE-Vio770, CD8-Brilliant Violet 510, CD38-VioBright FITC, CD45-APC-H7, CD45RA-VioGreen, CD45RO-VioBlue, FasL (CD178)-Vio Bright B515, CCR7-APC, granzyme B-APC, IFN-γ-PE, Ki-67-APC, PD-1-PE-Vio770, perforin-Pacific Blue, and Fixable Viability Dye eFluor 780 (Table S1).

## Expansion and enrichment of SC2-STs

$8 \times 10^7$ PBMCs were cultivated in 12-well culture plates (Corning) at a concentration of $5 \times 10^6$ cells/ml in RPMI 1640 medium, 5% (vol/vol) AB serum, 1% (vol/vol) penicillin/streptomycin, supplemented rhIL-2 (20 IU/ml), rhIL-7; (10 ng/ml), and rhIL-15 (10 ng/ml) (Miltenyi Biotec). The cells were stimulated with Peptivator SARS-CoV-2 Select (1 $\mu$g/ml) and cultured for 14 d, replenishing every 2–3 d with fresh supplemented media. At day 7, the cells were washed, counted, and split. As negative control, a 14-d expansion was carried out in 4 subjects (donors 8, 9, 11, and 12), using the same procedure without SARS-CoV-2 peptide stimulation to confirm the specificity of the methodology.

At day 14, the immunomagnetic separation of SARS-CoV-2-specific IFN-γ producing cells was carried out. Cells were re-stimulated with SARS-CoV-2 Peptivator and labelled with IFN-γ Secretion Assay Detection Kit as described before. After the addition of IFN-γ PE antibody, anti-PE Magnetic Microbeads (Miltenyi Biotec) were added and incubated for 15 min at 4°C. SC2-STs were isolated with MiniMACS Separator and MS Columns (Miltenyi Biotec). If the cell count after the first enrichment process was higher than $5 \times 10^6$, a second consecutive selection was performed using a new MS column. To determine the purity and phenotype of the expanded and isolated cells, they were stained with anti CD3-Violet Fluor 450, CD4-APC, CD8-PE-Vio770, CD45-FITC, and 7-AAD, and analyzed by flow cytometry (Table S1). CD56-APC-Vio770 was only determined in four subjects for the characterization of NK subpopulation in the isolated product. Because of stimulation-induced CD3 internalization, the CD3$^{low}$ population analysis was included in flow cytometry analysis.

## Cytotoxicity testing of SC2-STs through Calcein-release-assay

To determine SARS-CoV-2-specific cytotoxicity, autologous PHA blast cells were used as target cells. Briefly, cryopreserved PBMCs were thawed and cultured in 24-well culture plates (Corning) at a final concentration of $1 \times 10^6$ cells/ml with RPMI 1640 media, 5% (vol/vol) human AB serum, and 1% (vol/vol) penicillin/ streptomycin. These cells were stimulated with PHA (Sigma-

Aldrich) (3 µg/ml) and incubated for 24 h at 37°C, 5% (vol/vol) $CO_2$. rhIL-2 was then added at a concentration of 20 UI/ml. After 72 h, PHA blasts were stimulated overnight with SARS-CoV-2 peptides (1 µg/ml). Unstimulated blasts were used as negative control.

To validate the role of HLA molecules in SC2-ST-induced cytotoxicity, SARS-CoV-2-pulsed PHA blasts were blocked with 25 µg/ml of anti-human HLA-A, -B, -C, and anti-human HLA-DR, -DP, -DQ (Becton Dickinson) for 2 h using unblocked SARS-CoV-2-pulsed PHA blasts as a reference control. PHA blasts were then stained with 2 µM of Calcein-AM (Invitrogen) for 1 h.

PHA blasts (target cells) were cocultured with enriched SC2-STs (effector cells) at different target to effector ratios (1:10, 1:5, and 1:1) in clear round-bottom 96-well microplates with phenol red free RPMI 1640 media and 5% (vol/vol) human AB serum. In addition, fresh media, target cells alone, and target cells with 2% (vol/vol) Triton X-100 (Sigma-Aldrich) were included to determine the background fluorescence, spontaneous, and maximum Calcein release, respectively. The background fluorescence value of fresh media was extracted from all measurements. After 4 h of incubation, the microplate was centrifuged and supernatants were transferred to a black flat-bottom 96-well microplate (Corning) for fluorescence measurement. Sample acquisition was carried out with a Synergy H1 spectrophotometer (Biotek, Agilent) with an excitation and emission filter (490 and 520 nm, respectively). Virus-specific lysis was calculated as follows:

$$Specific\ lysis = \frac{Test\ release - spontaneous\ release}{Maximum\ release - spontaneous\ release} \times 100$$

### Proliferative potential analysis of SC2-STs based on CFSE assay

SARS-CoV-2-specific enriched cells were labelled with 5 µM CFSE (Sigma-Aldrich), followed by dye neutralization with human AB serum at 4°C. $7.5 \times 10^4$ CFSE–labelled cells were co-cultured for 5 d in round bottom 96-well microplates (Corning) with $7.5 \times 10^4$ autologous feeders, obtained from thawed PBMCs, previously loaded with Peptivator SARS-CoV-2 Select (1 µg/ml). Simultaneously, specific cells were cocultured with autologous untouched feeders as negative control or activated feeders pulsed with 250 ng/ml of PHA and rhIL-2 (200 IU/ml) as positive control. On day 5, samples were stained with CD3-Violet Fluor 450, CD4-APC, CD8-PE-Vio770, CD45-APC-H7, and 7-AAD (Table S1), and analyzed by flow cytometry methodology.

### Data analysis

GraphPad Prism 9.3.1 software and R 3.4.0 (R Foundation for Statistical Computing) were used for data plotting and statistical analysis. Flow cytometry data were analyzed using the FACSDiva 6.1.3 (Becton Dickinson) and FlowJo 9.9.6 (FlowJo) softwares. For flow cytometry analysis, the spillover spread associated with the calibration and compensation of the equipment was taken into consideration. Data were presented as mean ± SD, or median ± interquartile range for fold expansion and donor characteristics presentation. Absolute counts of SC2-STs and related subsets were calculated applying the proportions obtained by flow cytometry with the gating strategy shown in Fig S3 to the number of cells

counted after PBMCs isolation, considering viable CD45[+] cells as the parent lineage. Experimental conditions were compared by two-tailed paired parametric $t$ tests for quantitative variables and Chi square or Fisher's exact test for categorical variables. For multiple comparisons, two-way analysis of variance (ANOVA) with Tukey's multiple comparison tests was used. Multivariate analysis was performed by generalized multivariate regression. For the final model, only variables that resulted significantly in the backward stepwise variable selection strategy were maintained. The non-parametric Spearman correlation test was used for correlation analysis. For all comparisons, statistical significance was defined as *$P < 0.05$; **$P < 0.01$, ***$P < 0.001$, ****$P < 0.0001$. Adobe Illustrator 26.3.1 (Adobe Inc.) was used for figure preparation.

## Supplementary Information

## Acknowledgements

We would like to thank all participants who joined our study and the nurses responsible for blood extraction. This study was supported by the Department of University, Innovation and Digital Transformation of Government of Navarra, with the grant 0011-3597-2020-000006.

### Author Contributions

R Gil-Bescós: data curation, formal analysis, validation, investigation, methodology, writing—original draft, review, and editing, and collected experimental data.
A Ostiz: methodology, writing—review and editing, and collected experimental data.
S Zalba: writing—review and editing and patient recruitment.
I Tamayo: formal analysis, writing—review and editing, and statistical analysis.
E Bandrés: methodology and writing—review and editing.
E Rojas-de-Miguel: methodology, writing—review and editing, and collected experimental data.
M Redondo: writing—review and editing and patient recruitment.
A Zabalza: conceptualization, investigation, and writing—review and editing.
N Ramírez: conceptualization, resources, supervision, funding acquisition, validation, investigation, project administration, and writing—original draft, review, and editing.

### Conflict of Interest Statement

The authors declare that they have no conflict of interest.

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
