## [Reviewer comments · Life Science Alliance]

Life Science Alliance

Potency assessment of IFN γ -producing SARS-CoV-2 specific T cells from COVID-19 convalescent subjects

Rubén Gil-Bescós, Ainhoa Ostiz, Saioa Zalba, Ibai Tamayo, Eva Bandrés, Elvira Rojas-de-Miguel, Margarita Redondo, Amaya Zabalza, and Natalia Ramirez

DOI: <https://doi.org/10.26508/lsa.202201759>

Corresponding author(s): Natalia Ramirez, Navarrabiomed, Universidad Pública de Navarra (UPNA), Hospital Universitario de Navarra (HUN), IDISNA (Navarra's Health Research Institute)

Review Timeline:

Submission Date:	2022-10-07
Editorial Decision:	2022-11-07
Revision Received:	2023-01-19
Editorial Decision:	2023-02-14
Revision Received:	2023-03-06
Editorial Decision:	2023-03-06
Revision Received:	2023-03-07
Accepted:	2023-03-07

Scientific Editor: Novella Guidi

Transaction Report:

November 7, 2022

Re: Life Science Alliance manuscript #LSA-2022-01759-T

Dr. Natalia Ramírez
Oncohematology Research Group, Navarrabiomed, University Hospital of Navarra, Public University of Navarra, Navarra
Medical Research Institute (IdiSNA);
Oncohematology Research Group
Irunlarrea 3, Street
Pamplona, Navarra 31008
Spain

Dear Dr. Ramírez,

Thank you for submitting your manuscript entitled "Potency assessment of IFN- γ -producing SARS-CoV-2 specific T cells from COVID-19 convalescent subjects" to Life Science Alliance. The manuscript was assessed by expert reviewers, whose comments are appended to this letter. We invite you to submit a revised manuscript addressing the Reviewer comments.

Thank you for this interesting contribution to Life Science Alliance. We are looking forward to receiving your revised manuscript.

Sincerely,

B. MANUSCRIPT ORGANIZATION AND FORMATTING:

Reviewer #1 (Comments to the Authors (Required)):

This is an interesting paper and well-written throughout. It addresses an important point - the functional characterisation of COVID-specific T cells which relate to those used for therapy. In this case the generation of SARS-CoV-2 specific T cells and the assessment of phenotype, cytotoxicity and proliferative capacity is reviewed. The introduction is effective and clear and provides the appropriate context for the use of virus-specific T cells (SC2-ST in this case) as a treatment for COVID-19. The results describe the generation of SC2-STs from 9 convalescent donors using an alternative approach to other authors in this field. The SC2-ST are generated from PBMC then subsequently isolated by CCS, rather than isolation of virus-specific cells first and expansion in feeder cells as has been previously reported. The figures are all well-presented and effective. In particular Figure 1 is an excellent summary of the study design. However I have several issues with the data itself which would need to be addressed for this manuscript to be published:

- 1) The principal issue is that there is a wide panel of flow analysis performed on the initial cells from the donors, but this assessment is not carried over to the expanded product, so it is difficult to determine what the end product phenotype is except at a very basic CD3/4/8 level. The phenotypic analysis should be extended to the expanded product too and discussed appropriately, especially in the context of differentiation status (Tem / Tcm etc).
- 2) The purity of the SC2-CTs after CCS isolation is described as very high (which at 70-75% is not high purity and this should be amended in the text - it is enriched), but there is no discussion or assessment of the residual 30% of cells in the expanded product, and no determination of the effect of this population on the subsequent expansion / cytotoxicity experiments. There should be more characterisation of this population - is it NK / NKT?
- 3) The paper focuses on functional responses of the expanded cells, but the PHA-blast cytotoxicity experiment is marginal at best, with poor evidence of cytotoxicity. The authors correctly show that CD4 ratio does impact on the experiment, but the HLA blocking is inconclusive and would suggest that the assay is not particularly effective. The positive and negative data for this and the proliferation experiment should be included. It might make more sense to segregate the data into high CD8+ isolates (0-2) and low CD8 ratios (4-8) to separate out differences effectively.
- 4) The proliferation data is also unconvincing and needs the positive control to indicate what maximal proliferation would look like. The un-pulsed and peptide pulsed lines should be equalised so that it is easier to assess differences. The quantitative data suggests significant differences between the CD3 and CD8 pulsed / un-pulsed data but the error bars would suggest this is unlikely.
- 5) The patient data table is not clear in some places - what does reinfection mean in this context? Also the data in supplemental fig 2 is quite revealing in that the figures in the top line, numbers 1,3,4 and 5 all indicate there is a trend to reduction over time with high responses at low DPSO, but this is reversed by the two patients at 650 DPSO - is this likely to be reinfection / re-exposure? This should be addressed as your data contradicts general published results.
- 6) Supplementary figure 1 is very good and demonstrates an appropriate flow cytometry gating strategy. However an issue with the CD3 gating is that it should include more of the CD3lo population as CD3 is down-regulated by stimulation, and these CD3lo/neg T cells are often the strongest IFN γ producers - this can be seen in your figure, so you should widen this gate to ensure you are not underestimating the IFN γ response.
- 7) One point in the text that needs to be removed is to suggest that these cells are suitable as an adoptive therapy. They are experimental research products which can be used to assess function, but they are not prepared or manufactured to GMP-compliant standards and therefore cannot be described as equivalent to a cell therapy.
- 8) One minor point - line 446 should be 'split' not 'splited'.

Reviewer #2 (Comments to the Authors (Required)):

Summary:

In the manuscript titled "Potency assessment of IFN- γ -producing SARS-CoV-2 specific T cells from COVID-19 convalescent subjects" Gil-Bescos et. al. present data describing the phenotypical and functional characteristics of SARS-CoV-2 specific T cells (SC2-ST) isolated from 9 convalescent patients (12 patients screened). SC2-ST were isolated using immunomagnetic enrichment of IFN γ secreting cells post-viral peptide stimulation. Cytotoxic potential of the SC2-ST was assessed using Calcein-release assay using peptide-pulse autologous PHA blasts. Ultimately, the authors suggest that enrichment of IFN γ producing

SC2-ST from convalescent donors, may be a viable option for generating SC2-STs for third party, "off-the-shelf" clinical application, used to treat those with severe SARS-CoV-2 infection. Overall, the manuscript is well organized and written (few minor grammatical issues throughout). However, the claims are not sufficiently supported by the data provided and require further analysis. I have outlined a few major and minor concerns below:

Major Concerns:

- 1) Figure 4: No negative control data is mentioned or presented. It would benefit the reader to know the frequency of IFN γ -producing cells in the absence of SARS-CoV-2 peptide stimulation to demonstrate a peptide-specific response.
- 2) Figure 5: Also missing negative control data. Were cells cultured for 14 days in the absence of SARS-CoV-2 peptides? Such data, if present, would support a peptide-specific expansion and not just non-specific expansion elicited from the culture media and cytokines (IL-2, IL-7, and IL-15).
- 3) Figure 6: Overall, this data does not support the claims that "SC2-ST show antigen-specific cytotoxicity in vitro which depends upon...the target to effector ratio" (Line: 203) and "SC2-ST maintain their proliferation potential upon antigen presentation" (Line 218). Specific comments can be found below:
 - a. Figure 6A: Given the lack of significant reduction in specific lysis upon blocking HLA I/II, are the authors suggesting incomplete blocking of MHC-mediated killing via the HLA I/II antibodies or is the interpretation that lysis is occurring in an MHC-independent manner? In my opinion this analysis is incomplete and requires further investigation to support the claims.
 - b. Figure 6B: It appears as if there are only 8 datapoints in this figure whereas 9 donors have been compared throughout the manuscript. Please list the number of replicates compared in each experiment in the figure legend and explain why any values may have been excluded.
 - c. Figure 6C: The CFSE dilution assay does not display the traditional CFSE dilution histogram with distinct peaks for each round of replication - thus it is difficult to draw conclusions about the replicative capacity of the cells (especially without axis scales/labels). Additionally, the authors acknowledge that the isolation/enrichment process itself activates the specific populations and thus this may not be an appropriate assay to assess the antigen-specific proliferation. Further analysis is required to support these claims.
 - d. Figure 6D: Are the "Unpulsed" bars representative of the Day 1 or Day 5 Unpulsed cells displayed in Fig 6C?
- 4) Interestingly, the authors identify donors 3 and 5 as having significantly higher CD4/8 ratios and reduced cytotoxic potential. Table 1 reveals that donors 3 and 5 are the only donors who were unvaccinated at the time of extraction. This is potentially an important finding not discussed by the authors. It is possible that the CD4-dominated responses and reduced cytotoxic potential of donors 3 and 5 is due to the fact that they have only immunity from natural infection, compared to their counterparts who have natural infection in addition to 1-3 rounds of vaccination (hybrid immunity). Further analysis of this data would significantly improve the impact of this manuscript by providing important clinical/immunological effects of vaccination.

Minor Concerns:

- 1) Line 75: Clinical trial NCT04401410 has been terminated. Additionally, there are many clinical trials currently ongoing evaluating T cells for COVID. Please update.
- 2) Line 104: The claim that "primary infection of convalescent donors was caused by different virus variants" is not supported by the data as no confirmatory sequencing was completed. Since predicted variant was determined through epidemiologic data, stating the infection in donors was "likely" or "most probably" caused by different variants would be more appropriate.
- 3) Line 159: "After the 14-day expansion, the percentage of SC2-ST increased to 3.92%..." What was the starting frequency?
- 4) Line 163: I would argue all 3 subsets showed great variability between donors. This is subjective language and interpretation. I would recommend using objective comparisons to support such claims. i.e. "CD8+ T cells had the greatest variability between donors with a CV of X compared to y and z for CD3+ and CD4+ subsets."
- 5) Line: 165: If calling out individual donors (3 and 5), it would be helpful to have the individual donors identified in the figures. This would also be beneficial throughout Figure 5 as the reader could then compare fold expansion (Fig 5D) to CD4/CD8 ratios (Fig 5E) for each donor, and which donors underwent 1 vs 2 rounds of enrichment (Fig 5F-G).
- 6) Line 441: I do not see a Supplementary Table 1 in my materials.

Reviewer #3 (Comments to the Authors (Required)):

Gil-Bescós and co-authors perform an in-depth phenotypical and functional characteristics of interferon- γ (IFN- γ) producing-SARS-CoV-2 specific T cells to determine their potency. The study is well executed, has some elements of novelty including the study of peptide-specific cytolytic and proliferative responses after antigenic re-challenge when others in previous studies just measured as redout cytokine production. There are some suggested improvement and clarification required to improve readability:

1- IFN- γ production calculation and criteria should be better explain. It is unclear if in Fig. 3 data are plotted background subtracted or not. If they are not subtracted, they should. This apply to the other data shown in the study. There is no mention of background subtraction in the methods pertaining each technique, not in the data analysis section.

2-Figures:

Figure 1: As proliferation and cytotoxic assays are carried at day 18, that should be added.

Figure 2: can be a panel of Figure1.

Figure 3: Consider mirroring Fig. 4 format plotting CD4 and CD8 separately and not stacked.

3- On the point raised:

"It has been demonstrated that there is a T memory repertoire against spike in uninfected donors (Grifoni et al., 2020b, Ogbe et al., 2021), but these cells have reduced functionality and cytotoxic potential (Kim et al., 2021), and it is unlikely that these donors would generate an adequate number of SC2-STs to meet the T-lymphocyte donation criteria"

There are actually studies showing that these cells can expand (see eg. Swadling et al., 2021 Nature). Accordingly the fact that those have not been considered in this study should be put as a limitation of the study instead.

Reviewer #1 (Comments to the Authors (Required)):

This is an interesting paper and well-written throughout. It addresses an important point - the functional characterization of COVID-specific T cells which relate to those used for therapy. In this case the generation of SARS-CoV-2 specific T cells and the assessment of phenotype, cytotoxicity and proliferative capacity is reviewed. The introduction is effective and clear and provides the appropriate context for the use of virus-specific T cells (SC2-ST in this case) as a treatment for COVID-19. The results describe the generation of SC2-STs from 9 convalescent donors using an alternative approach to other authors in this field. The SC2-ST are generated from PBMC then subsequently isolated by CCS, rather than isolation of virus-specific cells first and expansion in feeder cells as has been previously reported. The figures are all well-presented and effective. In particular Figure 1 is an excellent summary of the study design. However, I have several issues with the data itself which would need to be addressed for this manuscript to be published:

1) The principal issue is that there is a wide panel of flow analysis performed on the initial cells from the donors, but this assessment is not carried over to the expanded product, so it is difficult to determine what the end product phenotype is except at a very basic CD3/4/8 level. The phenotypic analysis should be extended to the expanded product too and discussed appropriately, especially in the context of differentiation status (T_{EM} / T_{cm} etc).

We thank the reviewer for their compelling suggestion. Following the reviewer's recommendation, we have reproduced the phenotypical characterization of SC2-STs after a 14-day expansion and extended in relation to latter suggestions (comment number 2). We used the same protocol described in the Material and Methods section of the manuscript. Due to ethical concerns and the fact that the immunological status of the enrolled donors has probably changed since the initial study was performed, we used frozen samples from 4 donors (n = 4) that had a surplus of cells in the initial study. We were unable to perform the assay on more subjects due to lack of biological material.

We found that the vast majority of expanded SC2-STs presented a T_{EM} phenotype, with a significant up-regulation of CD38, PD-1 and Ki-67. Similar results were reported by *Basar et al., 2021 Cell Rep*, and the up-regulation of these markers demonstrates the activation capacity of SC2-STs after an antigenic re-challenge. Regarding cytotoxicity markers, granzyme B showed a significant reduction in its percentage. Although this may seem contradictory to the fact that T_{EM} phenotype has increased, previous studies showed that the absence of type I IFNs and IL-12 in culture media can lead to a decrease in the expression of this cytolytic molecule (Curtsinger et al, 2005; Nowacki et al, 2007). Perforin and FasL percentages were not significantly altered.

The results of this experimental assay and the statistical analysis have been included and discussed in the main text (lines 180 to 185; 311 to 315; 330 to 336; 354 to 357) and a new table (Table 2) has been added to the Results section.

2) The purity of the SC2-CTs after CCS isolation is described as very high (which at 70-75% is not high purity and this should be amended in the text - it is enriched), but there is no discussion or assessment of the residual 30% of cells in the expanded product, and no determination of the effect of this population on the subsequent expansion / cytotoxicity experiments. There should be more characterisation of this population - is it NK / NKT?

Following the reviewer's recommendation, we have performed a 14-day expansion and subsequent immunomagnetic isolation with the same 4 donors mentioned in the previous question, in order to characterize the residual IFN- γ negative population (residual cells) in the enriched product. We used the same protocol mentioned in the Material and Methods section, incorporating an anti-CD56 APC-Vio770 antibody to the flow cytometry panel (included in the main text in line 545 to 546). We obtained a similar enrichment to that reported in the main text, with a mean of $63.4\% \pm 11.9\%$ vs $75.5\% \pm 11.9\%$. Effectively, within the remaining 36%, we found a median of $18.13\% \pm 8.53\%$ CD56+ cells. Due to a complete CD3+ internalization in the enriched product, NK and NKT cells could not be differentiable. This CD3+ internalization did not hinder the analysis of the expanded product (prior to immunomagnetic enrichment), where there were on average 8 times more NK cells than NKT cells. Therefore, this ratio probably did not significantly change after enrichment. These results were included and discussed in the main text (lines 200 to 207 and 388 to 396). The effect of NK/NKT cells in the cytotoxicity is further analyzed in the next question.

Additionally, we have changed the text line in which we refer to the final product as pure (line 209).

3) The paper focuses on functional responses of the expanded cells, but the PHA-blast cytotoxicity experiment is marginal at best, with poor evidence of cytotoxicity. The authors correctly show that CD4 ratio does impact on the experiment, but the HLA blocking is inconclusive and would suggest that the assay is not particularly effective. The positive and negative data for this and the proliferation experiment should be included. It might make more sense to segregate the data into high CD8+ isolates (0-2) and low CD8 ratios (4-8) to separate out differences effectively.

In order to further clarify the effect size of the level of evidence of the underlying process, we have now performed a multivariate generalized mixed effects model accounting for the cell stimulation conditions, the effector to blast ratio and for all the rest of baseline variables.

In this sense we tried to estimate the effect adjusting for the following confounders: age, sex, DPSO, IFN- γ percentage after immunomagnetic isolation, the CD4+IFN- γ + / CD8+IFN- γ + ratio, the target to effector ratio and the treatment of the cell line (SARS, HLA-blocking, and negative control). The vaccination status could also be integrated into this analysis but since it showed strong association with CD4+IFN- γ + / CD8+IFN- γ + ratio, and only 2 patients were unvaccinated, statistical power was prioritized using the quantitative variable.

CD4+IFN- γ + / CD8+IFN- γ + ratio, the type of target cell stimulation, and the target to effector cell ratio were independently associated with the level of cytotoxicity. Moreover, using the SARS stimulated cells as reference, the effect of HLA blockade could be measured, showing a decrease in cytotoxicity level by 8.9 units (CI95% [-13.41; -4.54]), whereas the cytotoxicity of the negative control decreased 16.6 units (CI95% [-21.14; -12.15]).

Variable		Estimate (95%CI)	p-Value
(Intercept)		24.612 [17.090, 32.134]	<0.001
CD4+IFN- γ + / CD8+IFN- γ +		-2.931 [-5.190, -0.673]	0.012
Target Cell Stimulation	SARS	Reference	
	Negative	-16.646 [-21.138, -12.153]	<0.001
	HLA Block	-8.976 [-13.415, -4.538]	<0.001
Target/effector ratio		1.207 [0.710, 1.705]	<0.001

Using this new statistical approach, we understand that the calcein-release assay is effective in determining the cytotoxicity of SC2-STs. Similarly, we consider that the HLA blockade shows conclusive results, by significantly reducing the cytotoxicity using SARS blasts as a reference. The fact that the specific lysis in the HLA blocked condition is not equal to the obtained in the negative blasts may be due to HLA-independent cytotoxicity, caused by the presence of the NK cell population in the selected product (effector cells), as demonstrated in the previous question. This effect was further discussed in the main text (Lines 388 to 396).

We understand that the ANOVA tests that was previously used in the manuscript did not correctly reflect these results, and we have replaced it with the results of this multivariate analysis (Lines 221 to 233 and 239 to 241).

In regard to the positive and negative controls, we have included the positive control in the proliferation assay figure (Figure 6D). Regarding the cytotoxicity assay, the negative control corresponds to the unstimulated blasts condition, specified in the main text in line 567. On the

other hand, the positive control was performed with the detergent Triton X-100, but since the calcein concentration in supernatant of this condition was used for the calculation of the specific lysis (equation in line 573), we understand that it is not possible to incorporate it into the figures.

4) The proliferation data is also unconvincing and needs the positive control to indicate what maximal proliferation would look like. The un-pulsed and peptide pulsed lines should be equalised so that it is easier to assess differences. The quantitative data suggests significant differences between the CD3 and CD8 pulsed / un-pulsed data but the error bars would suggest this is unlikely.

As commented in the previous question, we have included the positive control values in the proliferation assay (Figure 6D). The reason for the apparent discrepancy between observed p-values and the error bars is that the error bars represent high variance between subjects, and the indicated p-values relate to the paired contrast (change in proliferation for each patient different from zero). The p-value shows that although patients showed heterogeneous initial MFI, in most of them a significant change in the MFI is observed in the same direction (lowering) comparing SARS-CoV-2 pulsed and unpulsed cells on day 5. This effect is observed in all donors in the CD3+ and CD8+ subsets, and in 7/8 donors in the CD4+ subset, as reflected in the main text (lines 255 to 257). We have included the statistical method used in this assay in the main text (lines 252 to 253) to avoid future misunderstandings.

5) The patient data table is not clear in some places - what does reinfection mean in this context? Also the data in supplemental fig 2 is quite revealing in that the figures in the top line, numbers 1,3,4 and 5 all indicate the there is a trend to reduction over time with high responses at low DPSO, but this is reversed by the two patients at 650 DPSO - is this likely to be reinfection / re-exposure? This should be addressed as your data contradicts general published results.

The reviewer raises a tantalizing question regarding reinfection and DPSO. In fact, we are planning to include this question into further research. However, the main problem with the current data is that we require a higher number of patients. Post-hoc selection of any 2 patients in a 9 patients sample could greatly change the results, so we tried to minimize the effect these extreme values have in the correlation making use of the nonparametric Spearman's correlation instead of Pearson's statistic. Nonetheless, we have included to study limitations the fact that undocumented asymptomatic reinfections were not taken into consideration (Lines 420 to 421). Additionally, we have included a small explanation in lines 111-112 and changed the patient data table (Table 1) to avoid confusion in the meaning of the term.

6) Supplementary figure 1 is very good and demonstrates an appropriate flow cytometry gating strategy. However an issue with the CD3 gating is that it should include more of the CD3lo population as CD3 is down-regulated by stimulation, and these CD3lo/neg T cells are often the strongest IFNg producers - this can be seen in your figure, so you should widen this gate to ensure you are not underestimating the IFNg response.

As commented by the reviewer, CD3 internalization is a well-documented phenomenon in activated T-cells and must be taken into consideration in any flow cytometry analysis of activated T-lymphocyte subpopulations. We were well aware of it, and it was taken into consideration in the analysis of flow cytometry data. We found differences between dot-plots on

day 0 and day 14, prior to immunomagnetic enrichment. While on day 0 some CD3⁺ internalization was observed (CD3^{low} subpopulation), it is on day 14 (after *in vitro* expansion and an additional peptide stimulation for isolation proceeding) when the positive and negative populations started to merge (CD3^{low} cells disappeared after entering the CD3 negative population). This phenomenon generates a dot plot in which it is difficult to establish the threshold for CD3⁺ analysis, and increases the risk of losing specific CD3⁻ cells in the gating. An example is shown below:

In both cases, we tried to include the CD3^{low} subpopulation in the gating strategy as far as possible. However, due to the low frequency of IFN- γ (particularly on day 0), we tried to be conservative in our gating strategy to avoid any CD3⁻ cell inclusion. We understand that with this gating method we may have lost some SC2-STs, but we consider that it is preferable to be strict in this analysis in order to avoid the analysis of non-specific cells that could significantly alter our results.

We have included in Material and Methods section an informative sentence to avoid any confusion about this phenomenon (lines 547 to 548).

7) One point in the text that needs to be removed is to suggest that these cells are suitable as an adoptive therapy. They are experimental research products which can be used to assess function, but they are not prepared or manufactured to GMP-compliant standards and therefore cannot be described as equivalent to a cell therapy.

As the reviewer indicates, the aim of our study is to validate the functionality and potency of SC2-STs as a hypothetical adoptive therapy, not its manufacture, since the assays were carried out under GLP conditions and not the GMP standards necessary for the production of any advanced therapy. We understand that some comments in the main text can be misleading, so they have been tinted (line 305 and 439-441) or deleted.

8) One minor point - line 446 should be 'split' not 'splited'.

As suggested by the reviewer, we have changed the grammar mistake in line 534.

Reviewer #2 (Comments to the Authors (Required)):

Summary:

In the manuscript titled "Potency assessment of IFN- γ -producing SARS-CoV-2 specific T cells from COVID-19 convalescent subjects" Gil-Bescos et. al. present data describing the phenotypical and functional characteristics of SARS-CoV-2 specific T cells (SC2-ST) isolated from 9 convalescent patients (12 patients screened). SC2-ST were isolated using immunomagnetic enrichment of IFN γ secreting cells post-viral peptide stimulation. Cytotoxic potential of the SC2-ST was assessed using Calcein-release assay using peptide-pulse autologous PHA blasts. Ultimately, the authors suggest that enrichment of IFN γ producing SC2-ST from convalescent donors, may be a viable option for generating SC2-STs for third party, "off-the-shelf" clinical application, used to treat those with severe SARS-CoV-2 infection. Overall, the manuscript is well organized and written (few minor grammatical issues throughout). However, the claims are not sufficiently supported by the data provided and require further analysis. I have outlined a few major and minor concerns below:

Major Concerns:

1) Figure 4: No negative control data is mentioned or presented. It would benefit the reader to know the frequency of IFN γ -producing cells in the absence of SARS-CoV-2 peptide stimulation to demonstrate a peptide-specific response.

As recommended by the reviewer, we have included an image of the IFN- γ production in the unstimulated control of the Intracellular Cytokine Staining to Figure 4. Additionally, we have included in the manuscript (Lines 134 to 136) the frequencies of IFN- γ producing CD3⁺ cells in the unstimulated and SARS-CoV-2-pulsed conditions in the Intracellular Cytokine Staining assay.

2) Figure 5: Also missing negative control data. Were cells cultured for 14 days in the absence of SARS-CoV-2 peptides? Such data, if present, would support a peptide-specific expansion and not just non-specific expansion elicited from the culture media and cytokines (IL-2, IL-7, and IL-15).

We thank the reviewer for this comment, as we consider it is a really important point not addressed in our manuscript. To answer the question, we performed a new expansion assay, using frozen cells from 4 donors in which an excess of cells was obtained during the initial study (n=4). Due to ethical concerns and the fact that the immunological status of the enrolled donors has probably changed since the initial study was performed, a new blood extraction procedure was not performed. The 14-day expansion assays were performed, following the same protocol used in the initial study. In this occasion, in addition to the expansion after SARS-CoV-2 peptide stimulation, we included another experimental condition in which cells were expanded with the homeostatic interleukins (rhIL-2, rhIL-7, rhIL-15) but without an antigenic stimulation, as a negative control of the expansion procedure. After 14 days, IFN- γ production was determined after antigen re-challenge. After expansion we observed a median percentage of CD3⁺IFN- γ ⁺ cells of 0.040% \pm 0.027% in the negative control, compared to 3.90% \pm 3.53% in the SARS peptide-stimulated condition. These results distinctly demonstrate

a SARS-CoV-2 specific expansion and not a proliferation directed by homeostatic cytokines. These results and the methodology used were included in the Results (lines 172-173) and Materials and Methods (lines 534-536) sections of the manuscript.

3) Figure 6: Overall, this data does not support the claims that "SC2-ST show antigen-specific cytotoxicity in vitro which depends upon...the target to effector ratio" (Line: 203) and "SC2-ST maintain their proliferation potential upon antigen presentation" (Line 218). Specific comments can be found below:

a. Figure 6A: Given the lack of significant reduction in specific lysis upon blocking HLA I/II, are the authors suggesting incomplete blocking of MHC-mediated killing via the HLA I/II antibodies or is the interpretation that lysis is occurring in an MHC-independent manner? In my opinion this analysis is incomplete and requires further investigation to support the claims.

To improve the analysis of the cytotoxicity assays results, we have performed a multivariate generalized mixed effects model to further analyze the effect size of different confounder variables, including age, sex, DPSO, IFN- γ percentage after immunomagnetic isolation, the CD4+IFN- γ + / CD8+IFN- γ + ratio, the target to effector ratio and the treatment of the cell line (SARS, HLA-blocking, and negative control).

Variable		Estimate (95%CI)	p-Value
(Intercept)		24.612 [17.090, 32.134]	<0.001
CD4+IFN- γ + / CD8+IFN- γ +		-2.931 [-5.190, -0.673]	0.012
Target Cell Stimulation	SARS	Reference	
	Negative	-16.646 [-21.138, -12.153]	<0.001
	HLA Block	-8.976 [-13.415, -4.538]	<0.001
Target/effector ratio		1.207 [0.710, 1.705]	<0.001

CD4+IFN- γ + / CD8+IFN- γ + ratio, the type of target cell stimulation, and the target to effector cell ratio were independently associated with the level of cytotoxicity. Moreover, using the SARS stimulated cells as reference, the effect of HLA blockade could be measured, showing a decrease in cytotoxicity level by 8.9 units (CI95% [-13.41; -4.54]), whereas the cytotoxicity of the negative control decreased 16.6 units (CI95% [-21.14; -12.15]). Although there is a significant decrease in cytotoxicity caused by HLA blockade, it is true that it is not equal to the negative control, so we hypothesize that this is caused by MHC-independent mechanisms. To prove this, we have also analyzed the presence of NK cells in the isolated product using the frozen material commented in the previous question (n=4). We used the same protocol mentioned in the Material and Methods, incorporating an anti-CD56 APC-Vio770 antibody to the flow cytometry panel (included in the main text in lines 545 to 546). We found a median of $18.13\% \pm 8.53\%$ CD56+ cells (due to CD3+ internalization, NK and NKT cells could not be differentiable) in the CD3+IFN- γ - population. These results were included in the main text (lines 200 to 207). NK cells exert cytotoxicity in a MHC-independent manner, and, since we

have demonstrated the presence of NK cells in our final product, we can theorize that the difference in the cytotoxicity between the HLA blocked condition and the negative control is probably caused by this NK population. We have included these thoughts in the main text (lines 388-396).

Additionally, we have changed the ANOVA tests that was previously used in the manuscript to the multivariate analysis, since we believe that it better explains the cytotoxicity results (Lines 221 to 233 and 239 to 241).

b. Figure 6B: It appears as if there are only 8 datapoints in this figure whereas 9 donors have been compared throughout the manuscript. Please list the number of replicates compared in each experiment in the figure legend and explain why any values may have been excluded.

In the functional assays (cytotoxicity and proliferation assays) we analysed 8 donors instead of 9, because the sample obtained from donor 4 did not lead to a sufficient number of SC2-ST after expansion and selection procedures, as described in lines 198-199. The number of replicates in each experiment is described in Figure 2. Nonetheless, we understand that it can be confusing and have included the number of replicates in each Figure legend to facilitate the reading and analysis of our results.

c. Figure 6C: The CFSE dilution assay does not display the traditional CFSE dilution histogram with distinct peaks for each round of replication - thus it is difficult to draw conclusions about the replicative capacity of the cells (especially without axis scales/labels). Additionally, the authors acknowledge that the isolation/enrichment process itself activates the specific populations and thus this may not be an appropriate assay to assess the antigen-specific proliferation. Further analysis is required to support these claims.

The reviewer is correct that the CFSE histogram does not present the distinct peaks that characterize this assay. However, we believe that despite this the proliferation capacity of SC2-STs is real, with a clear and significant change in the value of MFI of the CFSE histogram comparing day 1 with day 5.

It is true that other selection methods, like MHC multimer technologies, do not activate cells when antigen-specific cells are selected. To use the isolated antigen-specific cells with this methodology in a proliferation assay would probably generate a greater difference between the negative control and the SARS peptide-stimulated condition on day 5. Our methodological design is based on the selection and analysis of functionally active cells (and not total SARS-CoV-2 specific cells as multimer technology employs), so their activation is essential for the isolation and the fact that there is proliferation in the negative control is intrinsic to our methodology. On the other hand, although the differences in CFSE dilution (value of MFI) are small, the paired analysis demonstrates antigen-specific proliferation. Therefore, the proliferation potential of SC2-STs is well demonstrated and we respectfully disagree with the reviewer on the lack of support for our claims.

Lastly, we have reissued Figure 6C and included axis scales and labels to make the figure easier to interpret.

d. Figure 6D: Are the "Unpulsed" bars representative of the Day 1 or Day 5 Unpulsed cells displayed in Fig 6C?

We have changed the label of Figure 6D, now it indicates that the Unpulsed cells represent data from day 5.

4) Interestingly, the authors identify donors 3 and 5 as having significantly higher CD4/8 ratios and reduced cytotoxic potential. Table 1 reveals that donors 3 and 5 are the only donors who were unvaccinated at the time of extraction. This is potentially an important finding not discussed by the authors. It is possible that the CD4-dominated responses and reduced cytotoxic potential of donors 3 and 5 is due to the fact that they have only immunity from natural infection, compared to their counterparts who have natural infection in addition to 1-3 rounds of vaccination (hybrid immunity). Further analysis of this data would significantly improve the impact of this manuscript by providing important clinical/immunological effects of vaccination.

To deepen into the reviewer's comment, we have performed a Fisher Exact Test to characterize the association of vaccination with other variables. We found that vaccination is significantly associated with the CD4+IFN- γ + / CD8+IFN- γ + ratio. These findings were included to lines 194 to 198.

Variable	levels	Unvaccinated	Vaccinated	p-value
Sex	Female	1 (50.0)	5 (71.4)	1.000
	Male	1 (50.0)	2 (28.6)	
DPSO	Mean (SD)	157.5 (25.2)	315.1 (255.3)	0.464
Age	Mean (SD)	27.0 (5.1)	27.1 (3.5)	0.969
IFN- γ	Mean (SD)	76.4 (15.4)	75.2 (8.7)	0.906
CD4+IFN- γ + / CD8+IFN- γ + ratio	Mean (SD)	5.5 (1.5)	0.8 (0.5)	<0.001

Therefore, it is possible to speculate that vaccinated subjects have a greater cytotoxic capacity due to the expansion of CD8+ cells. On the contrary, unvaccinated donors, which do not present hybrid immunity, have a greater expansion of CD4+, which significantly reduces the cytotoxic potential.

We want to thank the reviewer for this interesting comment, and we are considering starting new assays to characterize the difference between natural and hybrid immunity in the future. However, having a small study group and only 2 unvaccinated donors, we consider to be difficult to draw firm conclusions on the subject. Even so, we have included new comments in the main text discussing this possibility (lines 381 to 386).

Minor Concerns:

1) Line 75: Clinical trial NCT04401410 has been terminated. Additionally, there are many clinical trials currently ongoing evaluating T cells for COVID. Please update.

We have eliminated clinical trial NCT04401410 from the main text and have included additional clinical trials to lines 82-83.

2) Line 104: The claim that "primary infection of convalescent donors was caused by different virus variants" is not supported by the data as no confirmatory sequencing was completed. Since predicted variant was determined through epidemiologic data, stating the infection in donors was "likely" or "most probably" caused by different variants would be more appropriate.

We have changed the indicated phrase in line 114, following the reviewer's suggestion.

3) Line 159: "After the 14-day expansion, the percentage of SC2-ST increased to 3.92%..." What was the starting frequency?

As suggested by the reviewer, we have included the percentage of SC2-STs at expansion start day in line 171.

4) Line 163: I would argue all 3 subsets showed great variability between donors. This is subjective language and interpretation. I would recommend using objective comparisons to support such claims. i.e. "CD8+ T cells had the greatest variability between donors with a CV of X compared to y and z for CD3+ and CD4+ subsets."

As recommended by the reviewer, we have included the CV for objective comparison of the *in vitro* expansion in the cellular subsets (lines 175-177).

5) Line: 165: If calling out individual donors (3 and 5), it would be helpful to have the individual donors identified in the figures. This would also be beneficial throughout Figure 5 as the reader could then compare fold expansion (Fig 5D) to CD4/CD8 ratios (Fig 5E) for each donor, and which donors underwent 1 vs 2 rounds of enrichment (Fig 5F-G).

We have included the donor ID in Figures 5C, E-G and 6B to facilitate the individual analysis of each donor as suggested by the reviewer. Identification was not included to Figure 5D since we consider that it would hinder the analysis of the figure due to the large number of points.

6) Line 441: I do not see a Supplementary Table 1 in my materials.

We apologize for our mistake during initial submission in which the Supplementary Materials section were incomplete. Supplementary Table 1 has been included in the Supplementary Materials section.

Reviewer #3 (Comments to the Authors (Required)):

Gil-Bescós and co-authors perform an in-depth phenotypical and functional characteristics of interferon- γ (IFN- γ) producing-SARS-CoV-2 specific T cells to determine their potency. The study is well executed, has some elements of novelty including the study of peptide-specific cytolytic and proliferative responses after antigenic re-challenge when others in previous studies just measured as redout cytokine production. There are some suggested improvement and clarification required to improve readability:

1- IFN- γ production calculation and criteria should be better explain. It is unclear if in Fig. 3 data are plotted background subtracted or not. If they are not subtracted, they should. This apply to the other data shown in the study. There is no mention of background subtraction in the methods pertaining each technique, not in the data analysis section.

To carry out the phenotypical and functional assays, donors had to meet these criteria:

- 1) IFN- γ -producing CD3+CD45+ cells percentage must be superior to 0.01%.
- 2) There must be at least 10 IFN- γ -producing CD3+ cells for every 1×10^5 CD45+ cells.
- 3) The CD3+IFN- γ + cell count after peptide stimulation must double the number of the unstimulated control.

It is true that a large majority of molecular and cellular biological techniques require the subtraction of the background. However, in our case the criteria provided by the manufacturer Miltenyi Biotec (manufacturing specialist of ATMPs) for the IFN- γ detection kit (based on flow cytometer) and selection of the donors were very clear. Specifically, the criterion 3 applies to the quantification of the CD3+IFN- γ + percentage in the unstimulated condition (background or negative control) indicating that background subtraction cannot be performed. Similarly, in the expansion and proliferation assays, we were interested in the comparison between the SARS-stimulated condition and the negative control to determine the specific effect of SARS-CoV-2 peptides. Lastly, regarding the Intracellular Cytokine Staining assays, it is not possible to differentiate the phenotype of background cells from specific cells, since both conditions are measured separately in cytometry, making it impossible to subtract the negative control. For this reason, we cannot make this background subtraction. However, we have included the negative control values from the Intracellular Cytokine Staining assays (lines 134-136 and Figure 4A) and from the *in vitro* expansion (Lines 172-173) to improve the visualization of our data.

Indeed, in the cytotoxicity assays, in which cytometry techniques were not used, background extraction was carried out measuring the background fluorescence of the culture media and extracting it from all fluorescence values. We have included an explanatory sentence in line 567 to explain said procedure in the manuscript.

2-Figures:

Figure 1: As proliferation and cytotoxic assays are carried at day 18, that should be added.

Functional proliferation and cytotoxicity assays were performed on day 15, not on day 18. We understand that the time marks in Figure 1 could have been misleading, so we have eliminated

those marks in which no experiment was carried out to avoid confusion when reading the Figure 1.

Figure 2: can be a panel of Figure 1.

We think that both figures represent different concepts and are important enough to go separately. While Figure 1 represents the experimental design, showing a summary of all the performed experiments; Figure 2 represents the methodological approach and the sample size changes due to inclusion criteria. Therefore, we respectfully disagree with the reviewer's comment on Figures 1 and 2. However if the reviewer considers this suggestion an important issue, we don't have any problem making the change.

Figure 3: Consider mirroring Fig. 4 format plotting CD4 and CD8 separately and not stacked.

Following the reviewer's recommendation, we have tested the separated format for Figure 4A:

We believe that the stacked figure is more suitable for visualizing our data, since it allows an easier analysis of the CD4+/CD8+ composition of each donor individually. Therefore, we respectfully disagree with the reviewer's comment on Figure 3.

3- On the point raised:

"It has been demonstrated that there is a T memory repertoire against spike in uninfected donors (Grifoni et al., 2020b, Ogbe et al., 2021), but these cells have reduced functionality and cytotoxic potential (Kim et al., 2021), and it is unlikely that these donors would generate an adequate number of SC2-STs to meet the T-lymphocyte donation criteria"

There are actually studies showing that these cells can expand (see eg. Swadling et al., 2021 Nature). Accordingly the fact that those have not been considered in this study should be put as a limitation of the study instead.

We thank the reviewer for the interesting comment and paper recommendation. Contrary to what we thought, the aforementioned paper demonstrates that SC2-STs (RTC-specific T cells) can be expanded *in vitro* and are capable of producing IFN- γ . Therefore, these subjects could be candidates to donate SC2-STs after an *in vitro* expansion, but their low IFN- γ production on day 0 makes them unsuitable for our study since they probably will not meet the previously established criteria. Nevertheless, we have included a section in the limitations of our study explaining that the expansion of SC2-STs from uninfected donors was not tested (Lines 418-420). Additionally, we have nuanced line 297 to include this approach in the Discussion.

February 14, 2023

Re: Life Science Alliance manuscript #LSA-2022-01759-TR

Dr. Natalia Ramirez
Navarrabiomed, Universidad Pública de Navarra (UPNA), Hospital Universitario de Navarra (HUN), IDISNA (Navarra's Health Research Institute)
Irunlarrea 3
Pamplona 31008
Spain

Dear Dr. Ramirez,

Thank you for submitting your revised manuscript entitled "Potency assessment of IFN γ -producing SARS-CoV-2 specific T cells from COVID-19 convalescent subjects" to Life Science Alliance. The manuscript has been seen by the original reviewers whose comments are appended below. While the reviewers continue to be overall positive about the work in terms of its suitability for Life Science Alliance, some important issues remain.

Our general policy is that papers are considered through only one revision cycle; however, given that the suggested changes are relatively minor, we are open to one additional short round of revision. Please note that I will expect to make a final decision without additional reviewer input upon resubmission.

Please submit the final revision within one month, along with a letter that includes a point by point response to the remaining reviewer comments.

To upload the revised version of your manuscript, please log in to your account: <https://lsa.msubmit.net/cgi-bin/main.plex>
You will be guided to complete the submission of your revised manuscript and to fill in all necessary information.

B. MANUSCRIPT ORGANIZATION AND FORMATTING:

Sincerely,

Reviewer #1 (Comments to the Authors (Required)):

The authors have provided a comprehensive response to the review comments and in general have undertaken sufficient effort

to address the issues raised throughout. We thank the authors for their efforts to respond to our earlier points.

The principal issue for me was the lack of effective phenotyping of the final product to the same standard that the initial isolated cells were done. The use of a sub-cohort of previously-frozen material for phenotypic analysis is reasonable, assuming that the thaw and analysis process was undertaken appropriately. Unfortunately, this has not been included in the materials and methods and therefore cannot be assessed. It is not clear in the text that these cells were previously frozen and then thawed and tested and this may have an impact on the resulting phenotype. This should be made clearer in the text (line 185) and the method of thaw and testing included in the M+M.

The results from this testing are very interesting, and it is odd that they are presented as a table rather than in graph format as shown in figure 4 - I would have preferred this to be presented in the same way so that it is easy for the reader to make a direct comparison with the start material data. The data presented in Table 2 indicates a huge skew to Tem cells with very high percentage of PD-1 / CD38 expression which the authors compare against results published by Basar et al. My interpretation of the expanded cells is that they are activated Tem but potentially senescent (high PD-1, low granzyme). Without corroborating analysis of LAG3 /Tim3 this cannot be confirmed, but the authors should acknowledge this in the discussion and it may explain the marginal results seen in the cytotoxicity assay.

I would also suggest that as a cell therapy this product would likely be very short-lived in the recipient and would potentially require multiple doses to maintain protection, whereas products with a higher Tcm compartment would be more persistent. This point should also be mentioned in the discussion (line 447).

The other corrections are sufficient and I think this paper should be published with the suggested amendments.

Reviewer #2 (Comments to the Authors (Required)):

Reviewer #2 (Comments to the Authors (Required)):

Summary:

In the manuscript titled "Potency assessment of IFN- γ -producing SARS-CoV-2 specific T cells from COVID-19 convalescent subjects" Gil-Bescos et. al. present data describing the phenotypical and functional characteristics of SARS-CoV-2 specific T cells (SC2-ST) isolated from 9 convalescent patients (12 patients screened). SC2-ST were isolated using immunomagnetic enrichment of IFN γ secreting cells post-viral peptide stimulation. Cytotoxic potential of the SC2-ST was assessed using Calcein-release assay using peptide-pulse autologous PHA blasts. Ultimately, the authors suggest that enrichment of IFN γ producing SC2-ST from convalescent donors, may be a viable option for generating SC2-STs for third party, "off-the-shelf" clinical application, used to treat those with severe SARS-CoV-2 infection. Overall, the manuscript is well organized and written (few minor grammatical issues throughout). However, the claims are not sufficiently supported by the data provided and require further analysis. I have outlined a few major and minor concerns below:

Major Concerns:

1) Figure 4: No negative control data is mentioned or presented. It would benefit the reader to know the frequency of IFN γ -producing cells in the absence of SARS-CoV-2 peptide stimulation to demonstrate a peptide-specific response. As recommended by the reviewer, we have included an image of the IFN- γ production in the unstimulated control of the Intracellular Cytokine Staining to Figure 4. Additionally, we have included in the manuscript (Lines 134 to 136) the frequencies of IFN- γ producing CD3+ cells in the unstimulated and SARS-CoV-2-pulsed conditions in the Intracellular Cytokine Staining assay. Reviewer Response: Inclusion of the flow plot with the Unstimulated cells is indeed beneficial. I still have concerns, however, as the gates depicted in Figure 4A are not consistent between the Unstimulated and the SARS-CoV-2 pulsed conditions (see image below). It appears that if the gate on the Unstimulated cells were lowered to match the stimulated cells, the % IFN γ + would be significantly increased, which would unsubstantiate the authors claims. Additionally, the frequencies included in the text (Lines 137-139) do not have any statistical analyses to objectively evaluate the different frequencies presented.

2) Figure 5: Also missing negative control data. Were cells cultured for 14 days in the absence of SARS-CoV-2 peptides? Such data, if present, would support a peptide-specific expansion and not just non-specific expansion elicited from the culture media and cytokines (IL-2, IL-7, and IL-15).

We thank the reviewer for this comment, as we consider it is a really important point not addressed in our manuscript. To answer the question, we performed a new expansion assay, using frozen cells from 4 donors in which an excess of cells was obtained during the initial study (n=4). Due to ethical concerns and the fact that the immunological status of the enrolled donors has probably changed since the initial study was performed, a new blood extraction procedure was not performed. The 14-day expansion assays were performed, following the same protocol used in the initial study. In this occasion, in addition to the expansion after SARS-CoV-2 peptide stimulation, we included another experimental condition in which cells were expanded with the homeostatic interleukins (rhIL-2, rhIL-7, rhIL-15) but without an antigenic stimulation, as a negative control of the expansion procedure. After 14 days, IFN- γ production was determined after antigen re-challenge. After expansion we observed a median percentage of CD3+IFN- γ + cells of 0.040% {plus minus} 0.027% in the negative control, compared to 3.90% {plus minus} 3.53% in the SARS peptide-stimulated condition. These results distinctly demonstrate a SARS-CoV-2 specific expansion and not a proliferation directed by homeostatic cytokines. These results and the methodology used were included in the Results (lines 172-173) and Materials and Methods (lines 534-536) sections of the manuscript.

Reviewer Response: This additional analysis is well done, given the limited sample available, and demonstrates a clear, SARS-CoV-2 specific expansion.

3) Figure 6: Overall, this data does not support the claims that "SC2-ST show antigen-specific cytotoxicity in vitro which depends

upon...the target to effector ratio" (Line: 203) and "SC2-ST maintain their proliferation potential upon antigen presentation" (Line 218). Specific comments can be found below:

a. Figure 6A: Given the lack of significant reduction in specific lysis upon blocking HLA I/II, are the authors suggesting incomplete blocking of MHC-mediated killing via the HLA I/II antibodies or is the interpretation that lysis is occurring in an MHC-independent manner? In my opinion this analysis is incomplete and requires further investigation to support the claims. To improve the analysis of the cytotoxicity assays results, we have performed a multivariate generalized mixed effects model to further analyze the effect size of different confounder variables, including age, sex, DPSO, IFN- γ percentage after immunomagnetic isolation, the CD4+IFN- γ /CD8+IFN- γ ratio, the target to effector ratio and the treatment of the cell line (SARS, HLA-blocking, and negative control).

Variable	Estimate (95%CI)	p-Value
(Intercept)	24.612 [17.090, 32.134]	<0.001
CD4+IFN- γ /CD8+IFN- γ	-2.931 [-5.190, -0.673]	0.012
Target Cell Stimulation SARS Reference		
Negative	-16.646 [-21.138, -12.153]	<0.001
HLA Block	-8.976 [-13.415, -4.538]	<0.001
Target/effector ratio	1.207 [0.710, 1.705]	<0.001

CD4+IFN- γ /CD8+IFN- γ ratio, the type of target cell stimulation, and the target to effector cell ratio were independently associated with the level of cytotoxicity. Moreover, using the SARS stimulated cells as reference, the effect of HLA blockade could be measured, showing a decrease in cytotoxicity level by 8.9 units (CI95% [-13.41; -4.54]), whereas the cytotoxicity of the negative control decreased 16.6 units (CI95% [-21.14; -12.15]). Although there is a significant decrease in cytotoxicity caused by HLA blockade, it is true that it is not equal to the negative control, so we hypothesize that this is caused by MHC-independent mechanisms. To prove this, we have also analyzed the presence of NK cells in the isolated product using the frozen material commented in the previous question (n=4). We used the same protocol mentioned in the Material and Methods, incorporating an anti-CD56 APC-Vio770 antibody to the flow cytometry panel (included in the main text in lines 545 to 546). We found a median of 18.13% {plus minus} 8.53% CD56+ cells (due to CD3+ internalization, NK and NKT cells could not be differentiable) in the CD3+IFN- γ population. These results were included in the main text (lines 200 to 207). NK cells exert cytotoxicity in a MHC-independent manner, and, since we have demonstrated the presence of NK cells in our final product, we can theorize that the difference in the cytotoxicity between the HLA blocked condition and the negative control is probably caused by this NK population. We have included these thoughts in the main text (lines 388-396).

Additionally, we have changed the ANOVA tests that was previously used in the manuscript to the multivariate analysis, since we believe that it better explains the cytotoxicity results (Lines 221 to 233 and 239 to 241).

Reviewer Response: First, I must admit I am not qualified to evaluate the appropriateness of the advance statistical analysis applied herein. The authors have done well to further characterize the final product and identify NK/NKT cells as a minor component. However, presence of NK/NKT cells in the final population alone does not prove that killing is mediated through said NK/NKT cells. Looking closely at Figure 6A, specific lysis for SARS Blasts at 1:10 is ~30%. Upon blocking MCH I/II, specific lysis at 1:10 is now reduced to ~24%, compared to specific lysis of Negative blasts at 1:10 which is ~10-12%. These data suggest that the majority of killing is therefore mediated through MHC-independent mechanisms which the authors attribute to the minor population of NK/NKT cells, and does not support the overall claim that SCT2-STs show antigen-specific cytotoxicity.

b. Figure 6B: It appears as if there are only 8 datapoints in this figure whereas 9 donors have been compared throughout the manuscript. Please list the number of replicates compared in each experiment in the figure legend and explain why any values may have been excluded.

In the functional assays (cytotoxicity and proliferation assays) we analysed 8 donors instead of 9, because the sample obtained from donor 4 did not lead to a sufficient number of SC2-ST after expansion and selection procedures, as described in lines 198-199. The number of replicates in each experiment is described in Figure 2. Nonetheless, we understand that it can be confusing and have included the number of replicates in each Figure legend to facilitate the reading and analysis of our results.

Reviewer Response: Thank you for clarifying.

c. Figure 6C: The CFSE dilution assay does not display the traditional CFSE dilution histogram with distinct peaks for each round of replication - thus it is difficult to draw conclusions about the replicative capacity of the cells (especially without axis scales/labels). Additionally, the authors acknowledge that the isolation/enrichment process itself activates the specific populations and thus this may not be an appropriate assay to assess the antigen-specific proliferation. Further analysis is required to support these claims.

The reviewer is correct that the CFSE histogram does not present the distinct peaks that characterize this assay. However, we believe that despite this the proliferation capacity of SC2-STs is real, with a clear and significant change in the value of MFI of the CFSE histogram comparing day 1 with day 5.

It is true that other selection methods, like MHC multimer technologies, do not activate cells when antigen-specific cells are selected. To use the isolated antigen-specific cells with this methodology in a proliferation assay would probably generate a greater difference between the negative control and the SARS peptide-stimulated condition on day 5. Our methodological design is based on the selection and analysis of functionally active cells (and not total SARS-CoV-2 specific cells as multimer technology employs), so their activation is essential for the isolation and the fact that there is proliferation in the negative control is intrinsic to our methodology. On the other hand, although the differences in CFSE dilution (value of MFI) are small, the paired analysis demonstrates antigen-specific proliferation. Therefore, the proliferation potential of SC2-STs is well demonstrated and we respectfully disagree with the reviewer on the lack of support for our claims.

Lastly, we have reissued Figure 6C and included axis scales and labels to make the figure easier to interpret.

Reviewer Response: I appreciate the additional information, and agree that given the selected approach, the data support the claims.

d. Figure 6D: Are the "Unpulsed" bars representative of the Day 1 or Day 5 Unpulsed cells displayed in Fig 6C?

We have changed the label of Figure 6D, now it indicates that the Unpulsed cells represent data from day 5.

Reviewer Response: Thank you for clarifying.

4) Interestingly, the authors identify donors 3 and 5 as having significantly higher CD4/8 ratios and reduced cytotoxic potential. Table 1 reveals that donors 3 and 5 are the only donors who were unvaccinated at the time of extraction. This is potentially an important finding not discussed by the authors. It is possible that the CD4-dominated responses and reduced cytotoxic potential of donors 3 and 5 is due to the fact that they have only immunity from natural infection, compared to their counterparts who have natural infection in addition to 1-3 rounds of vaccination (hybrid immunity). Further analysis of this data would significantly improve the impact of this manuscript by providing important clinical/immunological effects of vaccination.

To deepen into the reviewer's comment, we have performed a Fisher Exact Test to characterize the association of vaccination with other variables. We found that vaccination is significantly associated with the CD4+IFN- γ + / CD8+IFN- γ + ratio. These findings were included to lines 194 to 198.

Variable levels Unvaccinated Vaccinated p-value

Sex Female 1 (50.0) 5 (71.4) 1.000

Male 1 (50.0) 2 (28.6)

DPSO Mean (SD) 157.5 (25.2) 315.1 (255.3) 0.464

Age Mean (SD) 27.0 (5.1) 27.1 (3.5) 0.969

IFN- γ Mean (SD) 76.4 (15.4) 75.2 (8.7) 0.906

CD4+IFN- γ + / CD8+IFN- γ + ratio Mean (SD) 5.5 (1.5) 0.8 (0.5) <0.001

Therefore, it is possible to speculate that vaccinated subjects have a greater cytotoxic capacity due to the expansion of CD8+ cells. On the contrary, unvaccinated donors, which do not present hybrid immunity, have a greater expansion of CD4+, which significantly reduces the cytotoxic potential.

We want to thank the reviewer for this interesting comment, and we are considering starting new assays to characterize the difference between natural and hybrid immunity in the future. However, having a small study group and only 2 unvaccinated donors, we consider to be difficult to draw firm conclusions on the subject. Even so, we have included new comments in the main text discussing this possibility (lines 381 to 386).

Reviewer Response: I agree with the authors' interpretation and feel this additional information has enhanced the manuscript. Best of luck with the future studies.

Reviewer #1 (Comments to the Authors (Required)):

The authors have provided a comprehensive response to the review comments and in general have undertaken sufficient effort to address the issues raised throughout. We thank the authors for their efforts to respond to our earlier points. The principal issue for me was the lack of effective phenotyping of the final product to the same standard that the initial isolated cells were done. The use of a sub-cohort of previously-frozen material for phenotypic analysis is reasonable, assuming that the thaw and analysis process was undertaken appropriately. Unfortunately, this has not been included in the materials and methods and therefore cannot be assessed. It is not clear in the text that these cells were previously frozen and then thawed and tested and this may have an impact on the resulting phenotype. This should be made clearer in the text (line 185) and the method of thaw and testing included in the M+M.

As recommended, we have included the thawing method in the Materials and Methods (lines 480-483) and have slightly changed lines 181-182 to facilitate the understanding of the experimental design.

The results from this testing are very interesting, and it is odd that they are presented as a table rather than in graph format as shown in figure 4 - I would have preferred this to be presented in the same way so that it is easy for the reader to make a direct comparison with the start material data. The data presented in Table 2 indicates a huge skew to Tem cells with very high percentage of PD-1 / CD38 expression which the authors compare against results published by Basar et al. My interpretation of the expanded cells is that they are activated Tem but potentially senescent (high PD-1, low granzyme). Without corroborating analysis of LAG3 /Tim3 this cannot be confirmed, but the authors should acknowledge this in the discussion and it may explain the marginal results seen in the cytotoxicity assay.

Following the reviewer's recommendation, we have eliminated Table 2 and have included those results in Figure 4 to facilitate the interpretation of our data by the reader. We have also changed the legend of Figure 4 to describe these new changes. The frequencies on day 14 of the cytotoxicity and activation markers were incorporated to Figure 4C and the memory phenotype subpopulations were included on Figure 4D, which is now separated from Figure 4C. Since the vast majority of post-expansion cells present T_{EM} phenotype, we consider that a pie chart allows a better visualization of the presented data than a bar graph, and it avoids an overload of the figure. We have also included the p-values for the increase in T_{EM} in the main text (lines 183-184).

Regarding the possibility that SC2-STs are senescent, we have carried out a bibliographical search to discuss this probability. While it is true that senescent cells express high percentages of PD-1 (*Janelle, V et al., 2021*), we have found discrepancies in how senescence affects the expression of Granzyme B. *Zhang J et al. (2021)*, and *Song Y et al. (2018)* report an increase of Granzyme B in senescent T-cells, whereas *Yang OO et al. (2005)* showed a decrease of said marker in HIV-specific cytotoxic senescent T cells. Nonetheless, we consider that it is still a plausible theory that should be taken into account and have included it to the discussion to enrich the manuscript (Lines 362-363).

References:

Song, Y., Wang, B., Song, R., Hao, Y., Wang, D., Li, Y., Jiang, Y., Xu, L., Ma, Y., Zheng, H., Kong, Y., & Zeng, H. (2018). T-cell Immunoglobulin and ITIM Domain Contributes to CD8⁺ T-cell Immunosenescence. *Aging cell*, 17(2), e12716. <https://doi.org/10.1111/ace1.12716>

Yang, O. O., Lin, H., Dagarag, M., Ng, H. L., Effros, R. B., & Uittenbogaart, C. H. (2005). Decreased perforin and granzyme B expression in senescent HIV-1-specific cytotoxic T lymphocytes. *Virology*, 332(1), 16–19. <https://doi.org/10.1016/j.virol.2004.11.028>

Zhang, J., He, T., Xue, L., & Guo, H. (2021). Senescent T cells: a potential biomarker and target for cancer therapy. *EBioMedicine*, 68, 103409. <https://doi.org/10.1016/j.ebiom.2021.103409>

I would also suggest that as a cell therapy this product would likely be very short-lived in the recipient and would potentially require multiple doses to maintain protection, whereas products with a higher Tcm compartment would be more persistent. This point should also be mentioned in the discussion (line 447).

As recommended by the reviewer, we have included the possible effect of this T_{EM} phenotype on a cell therapy product based on SC2-ST (lines 316-318).

Reviewer #2 (Comments to the Authors (Required)):

Reviewer Response: Inclusion of the flow plot with the Unstimulated cells is indeed beneficial. I still have concerns, however, as the gates depicted in Figure 4A are not consistent between the Unstimulated and the SARS-CoV-2 pulsed conditions (see image below). It appears that if the gate on the Unstimulated cells were lowered to match the stimulated cells, the % IFN γ + would be significantly increased, which would unsubstantiate the authors claims. Additionally, the frequencies included in the text (Lines 137-139) do not have any statistical analyses to objectively evaluate the different frequencies presented.

We thank the reviewer for their thoughtful recommendations. It is true that the gating strategy in the unstimulated and the peptide-stimulated condition should match, as long as the analysis conditions are identical. However, this is not the case, since different fluorochrome-associated antibodies were used in the unstimulated and stimulated conditions.

Control samples were labeled with 5 fluorochrome-associated antibodies, whereas peptide-stimulated conditions were labeled with 8 fluorochrome-associated antibodies. Intracellular staining panels included a viability dye (Fixable Viability Dye eFluor™ 780). In both cases the same compensation was used for sample acquisition, which was automatically calculated by FACSDiva 6.1.3 Software (Becton Dickinson) using the 8 photomultipliers of the flow cytometer during the FACSCanto II Cytometer (Bacton Dickinson) calibration and compensation procedure. Thus, the compensation did not have the same effect in both cases, generating slightly different images. We initially considered using a compensation for each flow cytometry panel, but we decided to use one compensation for all panels for the sake of maintaining the same acquisition conditions and avoid greater variability inter-samples.

Specifically, the differences between the IFN- γ -PE dot plots is also explained by the Spillover Spread effect. This spread consists of an increase in the width of a positive population, which influence the images of other channels. It is linked to the aforementioned compensation, cannot

be eliminated and increases with higher staining intensities of the antibody. FITC is a fluorochrome that usually spills into the PE channel, and needs to be compensated. In the unstimulated control panel, we used CD45 FITC, an antibody that exhibits bright staining intensity due to the great expression density of CD45. On the contrary, in the peptide-stimulated conditions, antibodies such as CD38 VioBright FITC or FasL VioBright 515 were used, two antigens that present much lower percentage of positive cells and a dim staining intensity. This difference caused changes in the PE images, as seen in our image. Hereunder, we show an image that we believe explains this phenomenon well, obtained from Figure 5.4.16 from *Maciorowski, Z et al, 2017*:

Although in this example they used different fluorochromes, this same phenomenon can be applied to our panels. The FITC spread in the negative control is higher due to the bright intensity of the CD45, raising the threshold of the PE negative population during the gate analysis. This threshold is maintained throughout all dot plots that include the PE photomultiplier, including the PE vs SSC presented in Figure 4A.

We understand that this can be misleading so we have indicated this fact in the Data Analysis section of our manuscript (lines 597-598). In addition, we have detailed the antibodies used in each flow cytometry panel in Supplementary Table 1. We hope that this answer has cleared up any doubts regarding the established threshold for specific cell analysis.

Lastly, as suggested, we have included the p-value of the t-test comparison between unpulsed and pulsed SC2-STs in line 136.

References:

Maciorowski, Z., Chattopadhyay, P. K., & Jain, P. (2017). Basic Multicolor Flow Cytometry. Current protocols in immunology, 117, 5.4.1–5.4.38. <https://doi.org/10.1002/cpim.26>

Reviewer Response: First, I must admit I am not qualified to evaluate the appropriateness of the advance statistical analysis applied herein. The authors have done well to further characterize the final product and identify NK/NKT cells as a minor component. However, presence of NK/NKT cells in the final population alone does not prove that killing is mediated through said NK/NKT cells. Looking closely at Figure 6A, specific lysis for SARS

Blasts at 1:10 is ~30%. Upon blocking MCH I/II, specific lysis at 1:10 is now reduced to ~24%, compared to specific lysis of Negative blasts at 1:10 which is ~10-12%. These data suggest that the majority of killing is therefore mediated through MHC-independent mechanisms which the authors attribute to the minor population of NK/NKT cells, and does not support the overall claim that SCT2-STs show antigen-specific cytotoxicity.

To answer to this suggestion, we will explain the arguments that have led us to the conclusion that there is antigen-specific cytotoxicity exerted by SC2-STs, taking into consideration the issues expressed by the reviewer. Figure 6A represents an overall image of the assay, but it does not take into account the confounders that significantly modulate the cytotoxicity response (CD4+/CD8+ ratio, stimulation condition and target to effector ratio) and contribute to the observed variance between subjects. For this reason, the multivariate analysis is necessary in our study. We consider that the analysis of the data to which the reviewer refers to should be integrated with the results obtained from the multivariate analysis presented in lines 224-236.

Variable		Estimate (95%CI)	p-Value
Target Cell Stimulation	SARS	Reference	
	Negative	-16.646 [-21.138, -12.153]	<0.001
	HLA Block	-8.976 [-13.415, -4.538]	<0.001

With this in mind, **and taking into account factors that modulate the cytotoxicity**, we would like to highlight two facts:

- After MHC I/II blockage, the cytotoxicity decreases in 8.9 units using the peptide-stimulated condition (named SARS in the previous table) as reference, which corresponds to the SC2-STs specific cytotoxic function.
- Similarly, for the negative blasts condition (without antigen-specific stimulation), the cytotoxicity decreases in 16.6 units: 8.9 units are due to the MHC-dependent cell death, and the remaining 7.7 is caused by MHC-independent cell death.

Given what has been said, there is MHC-independent cell death, probably induced by the presence of NK cells detected in the final product. However, as the multivariate analysis reveals, the contribution of SC2-STs antigen-specific cytotoxicity remains superior to the MHC-independent cells, regardless of other significant confounders (which do influence the results shown in Figure 6). Therefore, based on the results of our study and after a rigorous statistical analysis, it is feasible to assert that antigen-dependent cytotoxicity by SC2-STs exists and that it is predominant in our final cellular product. Lastly, we have added new lines to the manuscript (lines 402-404) to highlight this fact and to improve the reasoning behind our conclusion.

March 6, 2023

RE: Life Science Alliance Manuscript #LSA-2022-01759-TRR

Dr. Natalia Ramirez

Navarrabiomed, Universidad Pública de Navarra (UPNA), Hospital Universitario de Navarra (HUN), IDISNA (Navarra's Health Research Institute)

Irunlarrea 3

Pamplona 31008

Spain

Dear Dr. Ramirez,

Thank you for submitting your revised manuscript entitled "Potency assessment of IFN γ -producing SARS-CoV-2 specific T cells from COVID-19 convalescent subjects". We would be happy to publish your paper in Life Science Alliance pending final revisions necessary to meet our formatting guidelines.

-please upload your table files as editable doc or excel files

Figure Check:

-Figure 5A needs more visible scale bars

A. FINAL FILES:

B. MANUSCRIPT ORGANIZATION AND FORMATTING:

Sincerely,

March 7, 2023

RE: Life Science Alliance Manuscript #LSA-2022-01759-TRRR

Dr. Natalia Ramirez
Navarrabiomed, Universidad Pública de Navarra (UPNA), Hospital Universitario de Navarra (HUN), IDISNA (Navarra's Health Research Institute)
Irunlarrea 3
Pamplona 31008
Spain

Dear Dr. Ramirez,

Thank you for submitting your Research Article entitled "Potency assessment of IFN γ -producing SARS-CoV-2 specific T cells from COVID-19 convalescent subjects". It is a pleasure to let you know that your manuscript is now accepted for publication in Life Science Alliance. Congratulations on this interesting work.

DISTRIBUTION OF MATERIALS:

Again, congratulations on a very nice paper. I hope you found the review process to be constructive and are pleased with how the manuscript was handled editorially. We look forward to future exciting submissions from your lab.

Sincerely,
